# Fermentation of Fruits and Vegetables: Bridging Traditional Wisdom and Modern Science for Food Preservation and Nutritional Value Improvements

**DOI:** 10.3390/foods14132155

**Published:** 2025-06-20

**Authors:** Prasad S. Gangakhedkar, Hemant W. Deshpande, Gréta Törős, Hassan El-Ramady, Tamer Elsakhawy, Neama Abdalla, Ayaz Shaikh, Béla Kovács, Rushikesh Mane, József Prokisch

**Affiliations:** 1Institute of Animal Science, Biotechnology and Nature Conservation, Faculty of Agricultural and Food Sciences and Environmental Management, University of Debrecen, Böszörményi Street 138, 4032 Debrecen, Hungary; toros.greta@agr.unideb.hu (G.T.); jprokisch@agr.unideb.hu (J.P.); 2Doctoral School of Animal Husbandry, University of Debrecen, Böszörményi Street 138, 4032 Debrecen, Hungary; 3Department of Food Microbiology and Safety, College of Food Technology, Vasantrao Naik Marathwada Agricultural University, Parbhani 431402, India; hemantd22@gmail.com; 4Soil and Water Department, Faculty of Agriculture, Kafrelsheikh University, Kafr El Sheikh 33516, Egypt; 5Agriculture Microbiology Research Department, Soil, Water and Environment Research Institute, Sakha Agricultural Research Station, Agriculture Research Center, Kafr El Sheikh 33717, Egypt; drelsakhawyg@gmail.com; 6Plant Biotechnology Department, Biotechnology Research Institute, National Research Centre, 33 El Buhouth St., Dokki, Giza 12622, Egypt; neama_ncr@yahoo.com; 7Institute of Food Science, University of Debrecen, Böszörményi út 138, 4032 Debrecen, Hungary; akovacsb@agr.unideb.hu (B.K.); 8MIT College of Food Technology, Chhatrapati Sambhajinagar 412201, India; manerp.foodtech@gmail.com

**Keywords:** anthocyanins, functional foods, food safety, fermentation, lactic acid bacteria, flavonoids, probiotics

## Abstract

Fermented fruits and vegetables are gaining increased attention due to their enhanced nutritional properties, extended shelf life, and potential health benefits. Driven by consumer demand for natural, plant-based, and functional foods, fermentation is emerging as a sustainable alternative to conventional preservation methods. This review highlights the role of lactic acid bacteria and other microorganisms in transforming fruit and vegetable substrates into probiotic-rich, bioactive foods. It explores traditional and emerging fermentation techniques, the influence of microbial consortia on product quality, and the impact of fermentation on antioxidant activity, gut health, immune modulation, and chronic disease prevention. Furthermore, the review addresses food safety concerns related to biogenic amines, nitrite accumulation, and microbial contamination, describing current solutions involving both conventional and non-thermal processing technologies. By synthesizing recent advances in microbial fermentation science and biotechnological innovations, this paper underscores the potential of fermented fruits and vegetables to contribute to functional food development, dietary diversity, and sustainable food systems.

## 1. Introduction

Fruit and vegetable products with advanced technological features and improved health advantages for consumers are currently highly popular among food industries. Fruits and vegetables are utilized as a substrate for fermentation because they contain a lot of bioactive compounds and vitamins, which may benefit health [1]. They are harvested fresh and, as such, have a shorter shelf life and are very perishable [2]. The majority of fruits and vegetables are eaten fresh or after minimal industrial processing, such as in canned form, dried, as juice, paste, salad dressings, sauces, and soups [3]. Fresh produce, especially that which has been minimally processed, has a limited shelf life due to its susceptibility to pathogen contamination and quick microbial deterioration [4]. Historically, food companies have depended on preservation methods to enhance food quality and prevent food loss [5]. To maintain the shelf life and quality preservation of food, several chemical preservatives are used; at present, these compounds are hazardous to both the environment and human health [6].

Fermentation is one of the traditional ways of preserving fruit and vegetable commodities. Eating fermented fruits and vegetables is a tradition in many parts of the world [7,8]. This method is commonly used around the world to preserve the nutritional value and improve the flavor of processed fruit and vegetable products. There are many different types of fermented fruits and vegetables, such as kimchi [9], paocai [10], and sauerkraut [11], which are common, that differ depending on the components, recipe, and methods of fermentation used. Cruciferous vegetables, such as cabbage, kale, mustard greens, or radishes, are the main ingredients used to make fermented vegetables. Carrot [12], ginger [13], cucumber [14], eggplant, beetroot [15], garlic [16], olive [17], and papaya [18] are some more common fruits and vegetables used in fermentation. Every day, a lot of fruit, such as oranges, apples, grapes, and pomegranates, are consumed in great quantities worldwide [19]. Pickled vegetables, such as cucumbers, radishes, carrots, nearly all types of vegetables, and even some green fruits, such as olives, papaya, and mango, are acid fermented in the presence of salt, worldwide [20,21]. Fruits and vegetables can be safely preserved using lactic acid fermentation. This improves the supply and availability of fruits and vegetable foods throughout the year and the nutritional benefits of foods for the consumer [22].

Fruits that are not processed through the use of refrigeration or pickling can be used in alcoholic fermentation: sugar from either fruits or fruits and vegetables is fermented into ethanol, and then forms acetic acid [23]. In light of the current interest on the matter and an existing gap in the literature, this paper’s primary objective is twofold: first, to underline the scientific evidence regarding the benefits of consuming fermented plant-based products; and second, to present all the nutritional aspects and enhancements that come from processing fruits and vegetables through the use of different state-of-the-art fermentation processes [24]. In conclusion, traditional processing techniques combined with modern innovation in regard to fermented fruits and vegetables can replace synthetic additives, which are generally no longer considered “natural” by consumers, while enhancing healthier food preparation schemes [25].

Since consumers have become aware of the nutritional value and health advantages of fruits and vegetables, their consumption has increased even more. Polyphenols, such as flavonoids, phenolic acids, and isoflavones, are abundant in these foods [26]. Furthermore, anthocyanins, flavonoids, and phenolic acids can be found in red fruits, such as blood oranges, strawberries, black chokeberries, bilberries, cherries, and sour cherries [27]. These bioactive substances exhibit cardiovascular-related preventive, anticancer, and antioxidant qualities [28].

The technique used in the continuous struggle against microorganisms that contaminate or pose a threat to food is known as food preservation. Certain GRAS (Generally Recognized as Safe) bacteria have been identified among these microorganisms as possible substitutes for perishable fruit and vegetable preservation [29]. Lactic acid bacteria (LAB) and/or their naturally occurring metabolites are examples of these types of microorganisms. LAB have historically been employed as preservatives due to their antibacterial qualities [30]. Moreover, LAB have important characteristics related to improved food safety and shelf life. Their ability to prevent pathogen growth, while synthesizing antimicrobial chemicals, such as hydrogen peroxide and diacetyl, is related to these advantages. In addition to fighting diseases through the preservation of nutrients, LAB also improves food safety [31]. Fermented fruits and vegetables are not only a good example of effective food preservation, but they also contribute to improved health through the production of beneficial compounds like phenols and vitamins [32].

*Limosilactobacillus fermentum*, *Lactiplantibacillus plantarum*, and *Levilactobacillus brevis* are the strains most frequently used in the industry for specific plant fermentation [33]. Fruits and vegetables can be kept longer due to the fermentation process, which is caused by lactic acid bacteria producing lactic acid and which decreases the pH as a result. A low pH inhibits the development of bacterial spores during food storage [34]. Food can, thus, be kept at low temperatures and without UV light for several months. However, because fermented foods require particular storage conditions, alternative ways to process them are currently being researched [35].

This review discusses the application of fermentation as a sustainable and effective technique for extending the longevity of fruits and vegetables, thereby mitigating post-harvest degradation and reducing food waste This study rigorously assesses the involvement of microorganisms, specifically lactic acid bacteria, in the bioconversion processes of plant-derived substrates, highlighting their role in augmenting nutritional value, enhancing the bioavailability of bioactive constituents, and the provision of extensive health advantages upon consumption. Moreover, this review presents insights into traditional fermented products made from fruits and vegetables across diverse cultural contexts, underscoring their historical significance and functional characteristics. Distinguishing itself from the existing literature, this review synthesizes recent progress in fermentation science, encompassing the employment of innovative microbial strains, co-culturing methodologies, and contemporary processing techniques, such as microencapsulation and precision fermentation. Furthermore, this review critically evaluates the potential hazards and safety concerns associated with fermentation practices and deliberates upon the current regulatory challenges and prospects. By harmonizing traditional wisdom with modern scientific advancements, this review provides an updated and multifaceted perspective on the fermentation of fruits and vegetables, with significant implications for functional food development, public health, and the transformation of sustainable food systems.

## 2. Methodology of the Review

A growing body of scientific literature has explored the nutritional enhancement and health-promoting properties of fermented fruits and vegetables, particularly in relation to the relevant bioactive compounds and safety profiles. To comprehensively examine this area, a structured methodology was adopted for collecting and synthesizing relevant studies. Initially, a detailed table of contents was outlined to define the scope of the review. Following this, a systematic literature search was conducted using reputable academic databases and publisher platforms, including ScienceDirect, Springer, PubMed, Frontiers, Wiley, Google Scholar, and Scopus. To ensure a broad yet focused coverage of the subject, multiple keyword combinations were used. These included: “fermented fruits and vegetables”, “lactic acid bacteria”, “bioactive compounds in fermented produce”, “health benefits of fermented foods”, “fermentation and food safety”, “fruit juice probiotics”, “LAB in plant-based fermentation”, “polyphenols in fermentation”, and “fermented plant-based functional foods”. The selection of the articles was based on several factors, including its publication in peer-reviewed journals, its relevance to the scope of the review, the journal impact factor, author credibility, and the recentness of the data. Preference was given to articles published within the last 10 years, with over 85% of the sources originating from the past five years, reflecting the latest advancements in fermentation technologies and microbial applications.

## 3. Biochemical Pathways in Fruit and Vegetable Fermentation

Fruit and vegetable fermentation is a microbial-driven biochemical process that converts carbohydrate-rich substrates into a diverse array of metabolites. Three main fermentation types exist, namely alcoholic, lactic acid, and acetic acid fermentation, each producing different end products through the action of distinct microbial species [36,37].

### 3.1. Alcoholic Fermentation

Alcoholic fermentation is a central metabolic pathway involved in fruit processing, primarily mediated by yeasts (e.g., *Saccharomyces cerevisiae*) and certain bacteria (e.g., *Zymomonas mobilis*). The process is initiated through glycolysis, wherein glucose is catabolized into pyruvate via either the Embden–Meyerhof–Parnas (EMP) pathway, especially when yeast is present, or the Entner–Doudoroff (ED) pathway (in regard to *Zymomonas* spp.). Pyruvate is subsequently decarboxylated into acetaldehyde by pyruvate decarboxylase (PDC), followed by NADH-dependent reduction to ethanol via alcohol dehydrogenase (ADH), thereby regenerating NAD^+^ to sustain glycolysis under anaerobic conditions [36,38].

The net stoichiometric equation is:C_6_H_12_O_6_→2C_2_H_5_OH + 2CO_2_

This pathway is particularly significant in regard to high-sugar fruits (e.g., apples, berries, persimmons), which serve as substrates for wine and other fermented beverages [36,37].

Recent studies highlight that phytosterol supplementation in grape must enhances fermentation kinetics and volatile ester production, improving wine aroma profiles [36].

### 3.2. Lactic Acid Fermentation

Lactic acid bacteria (LAB) drive lactic acid fermentation, which is subclassified into homolactic and heterolactic pathways, based on metabolic end products [39,40].

#### 3.2.1. Homolactic Fermentation

Homofermentative LAB (e.g., *Lactobacillus delbrueckii*, *Streptococcus thermophilus*) convert glucose exclusively into lactate via the EMP pathway. Pyruvate is reduced to L-lactate by lactate dehydrogenase (LDH), with concomitant NAD^+^ regeneration [41]:C_6_H_12_O_6_→2CH_3_CHOHCOOH

#### 3.2.2. Heterolactic Fermentation

Heterofermentative LAB (e.g., *Leuconostoc mesenteroides*, *Levilactobacillus brevis*) utilize the phosphoketolase pathway, yielding lactate, ethanol, and CO_2_ in equimolar ratios:C_6_H_12_O_6_→CH_3_CHOHCOOH + C_2_H_5_OH + CO_2_

This bifurcated pathway involves glucose-6-phosphate dehydrogenase (G6PD) and phosphoketolase, redirecting the carbon flux toward multiple end products [39,40].

Recent applications include fruit by-product valorization, wherein LAB fermentation of apple pomace and cocoa bean shells was found to enhance bioactive compounds and sensory profiles in novel yoghurt-style beverages [39].

### 3.3. Acetic Acid Fermentation

Acetic acid bacteria (AAB; *Acetobacter*, *Gluconobacter*) mediate the aerobic oxidation of ethanol into acetic acid, a key process in fruit vinegar production. The reaction proceeds via two enzymatic steps:Ethanol→Acetaldehyde (catalyzed by membrane-bound alcohol dehydrogenase, ADH);Acetaldehyde→Acetic acid (via aldehyde dehydrogenase, ALDH).

The overall exergonic reaction is:C_2_H_5_OH + O_2_→CH_3_COOH + H_2_O

Electrons are shuttled through the respiratory chain, coupling acetic acid synthesis with ATP generation via oxidative phosphorylation [37]. AAB exhibit acid tolerance through the following:Periplasmic localization of ADH/ALDH;Proton extrusion mechanisms;Membrane lipid adaptations.

Fruit vinegars (e.g., apple, berry, persimmon) are increasingly valorized as sustainable by-products of surplus fruit processing [37,41].

### 3.4. Secondary Metabolic Pathways

Beyond primary fermentation, secondary metabolic pathways, including esterification, ketogenesis, and oxidative side-reactions, generate a diverse array of volatile organic compounds (VOCs), such as esters (e.g., ethyl acetate, isoamyl acetate), higher alcohols (e.g., isobutanol, phenylethanol), and aldehydes (e.g., acetaldehyde, furfural), which critically shape the organoleptic properties of fermented products by contributing fruity, floral, alcoholic, or nutty sensory notes. Additionally, some acetic acid bacteria (AAB) strains exhibit metabolic flexibility by further metabolizing acetate via the tricarboxylic acid (TCA) cycle, leading to biphasic growth kinetics, characterized by an initial phase of rapid growth fueled by ethanol oxidation, followed by a slower phase, sustained by acetate assimilation. This dynamic metabolic activity, particularly observed in high-acid fermentations like vinegar production, highlights the complex interplay between microbial physiology and product quality in fermentation systems [42,43]. Suggested biochemical pathways involved in fruit fermentation are presented in Figure 1.

## 4. Current Market for Fermented Fruit and Vegetable Products

The tendency towards consumables that improve health and lower the chances of illness is rising among present-day health-conscious consumers, thereby promoting the advancement of the global functional foods market [44]. A considerable segment of the emerging functional food category is constituted by fermented foods derived from grains, legumes, root vegetables, fruits, and other types of vegetables [45]. At present, fermented fruit and vegetable products are gaining increasing significance due to the rising prevalence of lactose intolerance and consumer inclinations towards low-cholesterol alternatives, rendering dairy products less attractive to consumers [46]. This shift in dietary preferences has led to a surge in the production and consumption of plant-based fermented products, which are not only rich in probiotics, but also offer a variety of flavors and textures that appeal to diverse palates [47]. This trend is further supported by the health benefits attributed to these products, which enhance their appeal among consumers seeking nutritious options.

Probiotic bacteria play a beneficial role in the lactic fermentation of fruit and vegetable products. Interestingly, customers have become more attracted to probiotic foods that contain fruit ingredients [48]. Fruit and vegetable drinks appeal to the increasing consumer need for a smaller amount of sugar-filled products, particularly considering that the movement against sugar is causing an evolution in European food tastes toward flavors that are sour, bitter, or acidic [49]. Additionally, using fruits and vegetables as lactic fermentation substrates has the benefit of incorporating nutrients and flavors unique to each food variety, producing goods with unique physicochemical and sensory qualities that customers value [50]. Comparing beverage formulations derived from various fermented Brazilian fruits with or without whey added indicates that consumers prefer lactose-free fermented fruit and vegetable juices and that added dairy ingredients are undesirable [51]. This consumer preference aligns with the growing trend of utilizing non-dairy lactic acid fermentation products, which offer distinctive flavors and health benefits, without the drawbacks for those with a lactose intolerance [52]. This shift towards non-dairy alternatives highlights the increasing consumer demand for functional beverages that not only cater to dietary restrictions, but also provide health-promoting attributes and innovative flavors [53]. As the market evolves, producers will likely adopt innovative fermentation methods and ingredient blends to meet diverse consumer preferences, improving product quality, variety, and consumer well-being The continuous innovation in fermentation processes and ingredient sourcing will further enhance the quality and variety of these products, ultimately benefiting consumer health and satisfaction (Table 1).

## 5. Fermentation Process and Role of Microorganisms

The fermentation of food products involves LAB and various other microorganisms, including pathogens, yeast, and molds [66]. LAB exist naturally in food, particularly in fermented foods, and are crucial in almost all food fermentation processes (Table 2) [67]. These bacteria produce organic acids that can prolong the shelf life of fermented products and are naturally present in soil, water, manure, waste, and plants, as well as in mucous membranes, including those of human and animal intestines, the mouth, skin, urinary tract, and genitals, likely having a beneficial effect on these organs [68]. Since its discovery, LAB have been utilized due to its various advantages across several fields, including its usage as a starter in food and feed fermentation, pharmaceuticals, probiotics, and as biological control agents. Taxonomists classify these bacteria into two separate species: *Firmicutes* and *Actinobacteria* [69]. The phylum Firmicutes includes *Lactobacillus* (and its newly classified genera, such as *Lacticaseibacillus*, *Lactiplantibacillus*, *Limosilactobacillus*, and others), *Lactococcus*, *Leuconostoc*, *Oenococcus*, *Pediococcus*, *Streptococcus*, *Enterococcus*, *Tetragenococcus*, *Aerococcus*, *Carnobacterium*, *Weissella*, *Alloiococcus*, *Symbiobacterium*, and *Vagococcus*, whereas the phylum Actinobacteria includes Atopobium and *Bifidobacterium* [70].

Microbes associated with the production of fermented foods not only provide health advantages, but also secrete many metabolites, which enhance the organoleptic qualities of fermented food products [71]. Recent research on the antibacterial, antifungal, antiviral, and antioxidant properties of the plant-based fermented beverage ‘*Kombucha*’ states that the microbial metabolism of tea polyphenols mediates its antioxidant properties [72]

This review aims to cover the general aspects of fermentation involving fruits and vegetables, some factors affecting fruit fermentation production, and fruits’ health benefits and nutritional enhancements. In regard to this topic, microbial fermentation notably involves several stages, depending on the digestibility of the product, the origin of the raw materials, the labor intensity, and many other factors. The fermentation process includes microbial activities that differ from one food product to another, according to the desired end products. Indeed, the subsequent activity may consist of one or a combination of lactic acid fermentation, alcoholic fermentation, and acetic acid fermentation [73]. Specific yeast and bacteria are considered to be responsible for the characteristic naturally fermented fruit flavors of such products and the preservation effect, but they also have significance concerning the types of fruit used and other factors. Fruit fermentation is primarily caused by various microorganisms, because of the presence of glucose, maltose, lactose, and other fermentable oligosaccharides, including some unsaturated sugars [74]. Therefore, the type of microorganism varies according to the type of fruit used, the process, and possibly the geographical location of the process, as well as the desired product [75].

**Table 2 foods-14-02155-t002:** Lactic acid bacteria species and optimum fermentation conditions.

Species	Fruit and Vegetable Sources	Optimal Temperature (°C)	Optimal pH Range	Refs.
*Lactiplantibacillus pentosus*	Eggplants, Cucumbers	30–37 °C	5.0–6.0	[76]
*Lactobacillus rossiae*	Pineapple	25–30 °C	4.5–5.5	[77]
*Latilactobacillus curvatus*	Peppers	25–30 °C	4.0–6.0	[78]
*Lactobacillus paraplantarum*	Cabbages, Caper	30–37 °C	4.0–5.5	[79]
*Leuconostoc mesenteroides*	White Cabbages, Carrots	20–25 °C	4.5–6.0	[80]
*Weissella confusa*	Peppers, Blackberries, Papaya	25–30 °C	4.0–5.5	[81]
*Pediococcus pentosaceus*	Cherries, Cucumbers, Cabbages	25–30 °C	4.0–5.5	[82]
*Limosilactobacillus fermentum*	French Beans, Red Beets	30–37 °C	4.0–5.5	[83]
*Weissella soli*	Carrots	25–30 °C	4.0–6.0	[84]
*Lactiplantibacillus plantarum*	Tomatoes, Plums, Kiwi	30–37 °C	3.5–6.5	[85]
*Enterococcus faecium*	French Beans, Tomatoes	30–37 °C	6.0–7.5	[86]
*Levilactobacillus brevis*	Tomatoes, Melon pod	25–30 °C	4.5–6.5	[87]

### 5.1. Metabolites Produced by Commonly Used Fermentation Microorganisms

Fermentation, particularly lactic acid fermentation, is a complex biochemical process driven by a diverse consortium of microorganisms, predominantly lactic acid bacteria (LAB), yeasts, and molds. These microorganisms metabolize carbohydrates and other substrates present in fruits and vegetables, leading to the production of a wide array of primary and secondary metabolites. These metabolites are crucial to the characteristic flavor, aroma, texture, and nutritional profile of fermented products. They also contribute significantly to the preservation and safety of these foods by inhibiting the growth of spoilage and pathogenic microorganisms. LAB are the most prominent group of microorganisms involved in fruit and vegetable fermentation. They are Gram-positive, non-spore forming, and typically anaerobic or micro-aerophilic bacteria that produce lactic acid as the major end product of carbohydrate fermentation. Key genera include *Lactobacillus*, *Leuconostoc*, *Pediococcus*, *Enterococcus*, and *Streptococcus*. The specific metabolites produced by LAB depend on the bacterial strain, the substrate composition, and the environmental conditions, such as temperature, pH, and oxygen availability (Figure 2).

### 5.2. Key Metabolites from LAB Fermentation

LAB fermentation generates key metabolites that influence the flavor, safety, texture, and health benefits of foods. Lactic acid is the primary organic acid involved in fermentation, imparting sourness and antimicrobial effects, while acetic, propionic, and succinic acids add sharper notes and aid food preservation [88,89]. Bacteriocins like nisin enhance food safety by inhibiting pathogens, and exopolysaccharides (EPS) improve food texture and act as prebiotics [90,91]. LAB also synthesize B vitamins and vitamin K, enriches products with bioactive peptides (contributing to the umami taste and health benefits), and produces aroma compounds like diacetyl (buttery) and esters (fruity/floral) [25]. Additionally, certain strains produce GABA, providing potential neuroactive benefits, such as stress relief and blood pressure regulation [92,93]. These metabolites collectively enhance the sensory, nutritional, and functional properties of fermented foods [94].

### 5.3. Specific Metabolites Involved in Fruit Fermentation

Fruit fermentation often involves LAB, but yeasts can also play a significant role, especially in the initial stages or in specific fruit fermentations (e.g., wine, cider). The high sugar content in fruits influences the metabolic pathways and end products involved in fermentation. Fruit fermentation, often driven by LAB and yeasts due to the high sugar content, yields diverse metabolites that shape the flavor, aroma, and health benefits of such foods. Key organic acids like lactic, acetic, citric, and malic acid contribute to food tartness, while ethanol and other alcohols (e.g., propanol, butanol) arise from yeast or LAB activity, influencing the aroma of food [89,95]. Esters, such as ethyl acetate and isoamyl acetate, impart fruity and floral notes, and phenolic compounds are modified by LAB, enhancing their bioavailability and antioxidant properties [96]. These metabolites collectively define the sensory and functional qualities of fermented fruit products.

### 5.4. Specific Metabolites Involved in Vegetable Fermentation

Vegetable fermentation is predominantly driven by LAB, which thrive in the typically lower sugar and higher fiber environment of vegetables. The resulting metabolites contribute to the distinct sour and savory profiles of fermented vegetables like sauerkraut and kimchi. Key metabolites include lactic and acetic acids (contributing to sourness) [89], mannitol (adding sweetness) [90], and small amounts of other short-chain fatty acids (SCFAs) like propionic and butyric acid [97], which benefit gut health. Sulfur compounds in cruciferous vegetables create pungent notes, while fermentation enhances B vitamins, vitamin K, and bioactive peptides with antioxidant and anti-inflammatory properties [25,93]. These transformations not only define the sensory and nutritional qualities of fermented vegetables, but also underscore their health-promoting potential. In summary, the fermentation of fruits and vegetables by commonly used microorganisms, primarily LAB, results in a rich tapestry of metabolites. These compounds not only define the sensory attributes of the fermented products, but also contribute significantly to their nutritional value and health-promoting properties. By detailing these specific metabolites, we aim to provide a more comprehensive and scientifically rigorous understanding of the transformations that occur during fermentation.

## 6. Optimization of Fermentation Using the Newest Techniques and Equipment

Fermentation, a venerable method of food preservation and transformation, has progressed from conventional artisanal approaches to meticulously optimized scientific methodologies [98]. While traditional practices, such as the solar fermentation of grapes into wine or the spontaneous fermentation of tomatoes, are predominantly influenced by environmental conditions and indigenous microbiota, contemporary fermentation systems leverage sophisticated instruments, meticulously defined microbial cultures, and exacting process controls to ensure consistency, safety, and efficiency [99].

### 6.1. Modern Fermentation Techniques

Fermentation methodologies can be broadly divided into spontaneous fermentation, which harnesses the innate microflora of the raw substrate, and controlled fermentation, wherein specific starter cultures are introduced to guide the process (Figure 3) [27]. Controlled fermentations have become the preferred choice in industrial contexts, due to their predictability and quality assurance [100]. A further classification can be made between solid-state fermentation (SSF) and submerged fermentation (SMF) [101]. Solid-state fermentation (SSF) pertains to the cultivation of microorganisms on solid substrates devoid of free-flowing water, presenting benefits such as reduced energy requirements, enhanced product yield per unit volume, and applicability in regard to the valorization of agro-industrial by-products; however, it frequently encounters difficulties associated with scaling up and restricted process regulation [102]. Submerged fermentation (SMF), conversely, is conducted in liquid media, which facilitates improved regulation of environmental parameters and simplifies downstream processing; nevertheless, it generally necessitates greater energy consumption, produces increased wastewater, and may result in diminished product concentrations in regard to particular applications [103]. The choice between SSF and SMF ultimately depends on the specific requirements of the fermentation process, including the type of microorganism used, the desired product, and economic considerations. SSF is predominantly utilized in traditional or small-scale operations, whereas SMF is prevalent in liquid food systems and large-scale production [104]. Cutting-edge technologies are profoundly revolutionizing fermentation methodologies, such as the use of immobilized cell systems, membrane bioreactors, and automated bioreactors, featuring real-time monitoring functionalities [105].

#### 6.1.1. Categorization of Modern Fermentation Techniques

Modern fermentation techniques can be broadly categorized based on several criteria, including the type of microbial involvement, the physical state of the substrate, and the operational mode. For the purpose of discussing fruit and vegetable fermentation, the most relevant categorizations often revolve around the presence or absence of starter cultures and the nature of the fermentation environment. The most common types of fermentation are as follows:(1)Spontaneous Fermentation

Spontaneous fermentation relies on the indigenous microbiota naturally present in the raw materials (fruits and vegetables) and in the processing environment. This traditional method is often employed in household and small-scale production, yielding products with unique regional characteristics. Spontaneous fermentation relies on indigenous microbes present in the raw materials and in the environment, making it simple, cost effective, and ideal for traditional small-scale production, often yielding unique regional flavors and preserving microbial biodiversity [106,107]. However, its uncontrolled nature leads to inconsistent quality, longer fermentation times, and higher risks of spoilage or contamination by undesirable microbes, raising potential safety concerns, despite the natural acidification caused by lactic acid bacteria that often helps suppress pathogens [108,87].

(2)Starter Culture Fermentation

Starter culture fermentation involves the deliberate inoculation of raw materials with known, characterized microorganisms (starter cultures) to initiate and guide the fermentation process. This method is widely adopted in industrial settings, due to the enhanced level of control offered and its reproducibility. Starter culture fermentation uses carefully selected, well-defined microbial strains to initiate and control the fermentation process, ensuring consistency, safety, and efficiency, making it ideal for industrial-scale production. Key advantages include precise control over the fermentation kinetics, leading to uniform product quality and faster processing times, as well as enhanced safety through the competitive exclusion of pathogens and the production of antimicrobial compounds. Additionally, starter cultures can be tailored for specific functional benefits, such as optimized flavor, texture, or health-promoting properties (e.g., probiotics, vitamins) [90,109]. However, this method has drawbacks, including higher production costs due to culture preparation and maintenance, reduced microbial diversity, which may limit flavor complexity, and potential sensory uniformity across different products, as the reliance on standardized strains can diminish regional uniqueness. Despite these trade-offs, starter culture fermentation remains essential for ensuring reliability and scalability in modern food fermentation [92,96].

(3)Solid-State Fermentation (SSF)

Solid-state fermentation involves the growth of microorganisms on solid substrates in the absence or near absence of free water. This technique is commonly used for producing enzymes, antibiotics, and some traditional fermented foods, wherein the substrate provides both nutrients and physical support. The advantages of this kind of fermentation are higher volumetric productivity [110], lower energy consumption [111], simpler downstream processing [112], and it mimics natural habitats [113]. The disadvantages are heat and mass transfer limitations [114], scale-up challenges [115], and contamination risks [116].

(4)Submerged Fermentation (SMF)

Submerged fermentation involves the growth of microorganisms in a liquid medium containing dissolved nutrients. This is the most widely used fermentation technique in industrial biotechnology for producing biomass, metabolites, and enzymes, due to its ease of control and scalability. Key advantages include excellent control and monitoring [117], ease of scale-up [118], efficient heat and mass transfer [119], and lower labor costs [120]. However, limitations include high energy demands for mixing and aeration, costly product recovery from dilute broths, and substantial water requirements [121,122,123].

(5)Precision Fermentation

Precision fermentation (PF) is a revolutionary technology that leverages microorganisms (bacteria, yeast, fungi) as microbial cell factories to produce specific functional ingredients, such as proteins, enzymes, fats, and flavor compounds, with high purity and efficiency. Unlike traditional fermentation, which focuses on bulk product transformation, PF is about producing specific molecules. This technology is gaining significant traction in the food industry for creating sustainable and animal-free alternatives to traditional ingredients. Key advantages include sustainability, customization and purity, scalability, and ethical considerations. However, the limitations include the cost of production, regulatory hurdles, and consumer acceptance [124,125].

(6)Continuous Fermentation

Continuous fermentation involves the continuous supply of fresh medium to the bioreactor and the continuous removal of fermented broth, maintaining the microorganisms in a state of constant growth and productivity. This contrasts with batch fermentation, whereby the process is run in discrete cycles. Concerning the mechanism, in regard to a continuous system, nutrients are continuously fed into the bioreactor, and an equal volume of fermented broth, containing cells and products, is simultaneously withdrawn. This maintains a steady-state environment, allowing for prolonged operation and higher overall productivity. Key advantages include higher productivity, consistent product quality, reduced downtime, and smaller reactor volume, whereas the limitations include contamination risks, process control complexity, and variable strain stability [117,126].

(7)Artificial Intelligence (AI) and Machine Learning (ML) in Regard to Fermentation

AI and ML are increasingly being integrated into fermentation processes to optimize the relevant parameters, predict outcomes, and enhance efficiency. These technologies enable data-driven decision making and automation, leading to more robust and productive fermentation systems. Key advantages are enhanced efficiency and yield, reduced development time, and improved reproducibility, whereas the limitations include high data requirements, model interpretability issues, and integration complexity [127,128]. These cutting-edge technologies represent the future of fermentation, enabling the production of novel, sustainable, and highly functional food ingredients and products. By incorporating these detailed discussions, we aim to provide a comprehensive and forward-looking perspective on modern fermentation techniques.

#### 6.1.2. Factors Influencing the Choice of Fermentation Method

The selection of an appropriate fermentation method for fruits and vegetables is a critical decision influenced by a multitude of factors, ranging from the desired product characteristics to economic and regulatory considerations. Understanding these factors is essential for optimizing the fermentation process and achieving the desired outcomes. The choice of fermentation method for fruits and vegetables depends on multiple factors, such as the raw material’s properties (sugar content, pH), determining the most suitable microbes (yeast for fruits, LAB for vegetables) [129]; the desired product traits (artisanal flavors favoring spontaneous fermentation, probiotics requiring starter cultures) [107]; the microbial requirements (LAB/yeasts/molds dictating the use of SSF or submerged methods) [113]; the production scale (small scale flexibility vs. industrial automation) [118]; costs (spontaneous being economical, controlled methods being more expensive) [126]; safety regulations (preferring controlled fermentation) [122]; the environmental impact (SSF’s lower resource use) [111]; and the available infrastructure/expertise [119]. Ongoing biotech advances continue to expand the fermentation-related possibilities.

### 6.2. Equipment for Optimized Fermentation

The optimization of fermentation processes is intrinsically associated with the equipment employed to sustain the appropriate conditions. The following tools and configurations are currently embraced:(1)Fermentation chambers or bioreactors, featuring integrated controls for temperature, pH, and dissolved oxygen levels [130];(2)Airlocks and anaerobic lids are designed to regulate gas exchange and reduce oxygen infiltration [131];(3)Glass or stainless-steel weights are utilized to maintain solids submerged within brine [132];(4)Sensor-based monitoring systems, including pH meters, thermometers, and automated stirring mechanisms [133];(5)Cold chain apparatus for post-fermentation storage, which is especially vital for the stability of probiotic products [134].

### 6.3. Microbial Interactions During Fermentation

The efficacy of fermentation is contingent upon the interactions among microbial consortia, predominantly lactic acid bacteria and yeasts, and, occasionally, spore-forming bacteria, such as *Bacillus* spp. [135] (Figure 4). These interactions are characterized by the following aspects:(1)Synergism, exemplified by co-fermentation involving LAB and yeasts that augments food flavor and shelf life [136];(2)Competitive exclusion, where acid-producing microorganisms inhibit the proliferation of spoilage organisms [137];(3)Successional colonization, according to which the microbial dominance shifts in response to changing environmental conditions [138].

**Figure 4 foods-14-02155-f004:**
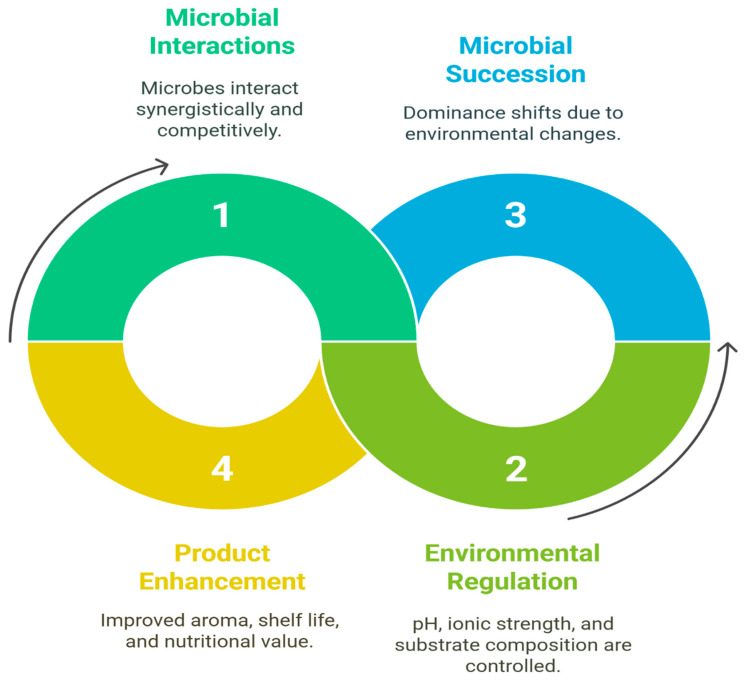
Microbial interactions and ecological dynamics during fermentation.

Understanding these interactions permits scholars and producers to systematically modify relevant variables, including the pH, ionic strength, and substrate composition, to enhance advantageous microbial activity [139].

## 7. Health Benefits of Fermented Fruits and Vegetable Products

Beneficial microbes developed during the pickling process are largely responsible for the health benefits of fermented fruits and vegetables [140]. These products exhibit antimicrobial, anticancer, and immune-boosting properties, and may help manage constipation, chronic illnesses, and irritable bowel syndrome. Regular consumption may also have long-term health implications [141].

### 7.1. Probiotics

Probiotics are live microorganisms that support human and animal health by maintaining the gut’s microbial balance. They are commonly introduced via starter cultures into fermented foods [57], primarily from *Lactobacillus* and *Bifidobacterium*, but also from Bacillus, Pediococcus, and yeasts, such as *Saccharomyces cerevisiae* and *S. boulardii* [142]. *Lactobacillus*, a diverse LAB group, is widely used in food fermentation and probiotic development. These Gram-positive bacteria produce lactic acid and promote health by restoring the gut microbiome, preventing infections, reducing allergies, and lowering cholesterol [143]. Probiotics may also inhibit pathogens, compete for adhesion sites, and modulate the immune response, helping prevent or manage conditions such as allergies, urogenital and digestive tract infections, irritable bowel syndrome (IBS), Inflammatory Bowel Disease (IBD), lactose intolerance, antibiotic-associated diarrhea, *Helicobacter pylori* gastritis, cystic fibrosis, colorectal cancer, and other cancers [144].

### 7.2. Antimicrobial Activity

A major benefit of fermented fruit and vegetable products is their ability to suppress pathogenic microorganisms, largely due to the presence of probiotics and their bioactive metabolites, such as peroxides and antimicrobial peptides. These compounds exert antimicrobial effects through various mechanisms, including nutrient competition, the disruption of cell membranes, the inhibition of DNA replication, interference with energy metabolism, the disruption of quorum sensing, and the prevention of biofilm formation [145]. Key antimicrobial substances in fermented products include bacteriocins, phenolic compounds, and extracellular polysaccharides. *Helicobacter pylori* (*H. pylori*), a major contributor to gastric cancer, can be inhibited by extracellular polysaccharides produced by *Lactobacillus* sp. PW-7, isolated from fermented pickles [146]. Dietary intake of kimchi has also been shown to suppress *H. pylori* proliferation in C57BL/6 mice [147]. Additionally, Pediococcus pentosaceus, a Gram-positive bacterium isolated from kimchi, exhibits anti-Listeria activity, attributed to the LysM structural domain of its antibacterial protein [148]. Probiotic strains in fermented vegetables can also synthesize a wide range of antimicrobial peptides, including nisin, which is approved by the U.S. Food and Drug Administration (FDA) for use as a food preservative. Ongoing research continues to identify novel peptides with promising antibacterial potential in fermented fruits and vegetables [149]. For example, Lactiplantibacillus plantarum CXG9, isolated from pickles, produces LD-phenylacetic acid, which has notable antimicrobial properties. Furthermore, phenolic compounds found in sauerkraut, such as 2,6-dihydroxyacetophenone (DHAP), 4-hydroxybenzaldehyde (HBA), and 4-hydroxyphenylethanol (4-HPEA), have shown variable antibacterial activity against foodborne pathogens [150].

### 7.3. Enhancing Intestinal Health

The consumption of fermented fruit and vegetable products can enhance the diversity of the gut microbiota (Figure 5). Most fermented fruits and vegetables contain a high amount of lactic acid bacteria; they are also a source of essential dietary fiber and several micronutrients [151]. In addition, they contain chemicals like glucosinolate and glutamine, which are potent antioxidants. They also contain abundant short-chain fatty acids, which prevent the expansion of dangerous bacteria and favor healthy ones. Oligosaccharides also form part of the regulation of intestinal microbiota, such as fructooligosaccharides, melanoidins, and some dietary phenolic compounds. It was seen that when fruits and vegetables were fermented for six months and given to people, there was an improvement in the imbalance in the gut; furthermore, when fruit and vegetables (which have undergone lactic acid fermentation for some time) are consumed, consumers show fewer symptoms of IBS [152]. They may also contribute to relieving constipation. The increased amount of fecal pollutants in the intestines increases the risk of intestinal disorders, and constipation reduces a person’s quality of life. The probiotics found in fermented fruits and vegetable products help in regard to being effective general constipation treatments [153].

### 7.4. Anticancer

The raw pickling materials used determine the anticancer properties of fermented fruits and vegetables, such as fermented cabbage. Studies have indicated that cruciferous vegetables subjected to fermentation possess anticancer properties [154]. The effect possibly pertains to the cruciferous vegetables’ resistance against cancer after being subjected to pickling, as indicated by in vitro experiments. Various studies have indicated that kimchi could potentially inhibit the proliferation of HT-29 colon cancer cells [155]. The fermentation of leaf mustard (EFeLM, extract of fermented leaf mustard) increased the presence of amino nitrogen and polyphenols, while reducing sugars and glucosinolates. The use of liquid chromatography–mass spectrometry (LC-MS) and gas chromatography–mass spectrometry (GC-MS) identified 117 bioactive compounds, with EFeLM showing a stronger inhibition in regard to HCT116 colon cancer cells than EFrLM (extract of fresh leaf mustard). EFeLM selectively induced apoptosis (via caspase-3 activation) and cell cycle arrest (cyclin B/cyclin D1/cyclin E downregulation) in cancer cells, sparing normal CCD-18Co colon myofibroblast cells. Polyphenols and/or fermentation-derived metabolites likely drive this effect, although exact compounds require isolation [156]. According to different research, consuming fermented cruciferous vegetables can reduce the risk of breast cancer, according to a study of 131 case-control studies concerning the disease among immigrants who had moved from Poland, residing in Cook County and the Detroit Metropolitan Area [157]. The findings indicated that the intake of raw or briefly steamed fermented cabbage reduces the incidence of breast cancer; however, in the case of cooked pickles, such incidences were not found, due to the destruction of probiotics or other invaluable ingredients by heat treatment [158].

### 7.5. Diabetes Inhibition

Type 2 diabetes is also mainly a consequence of a decrease in carbohydrate metabolism. Glucosidase inhibitors can reduce blood sugar levels by delaying the conversion of carbohydrates into glucose [159]. Because DPP-IV is capable of degrading the GLP-1 and GIP-two peptides capable of stimulating the pancreatic secretion of insulin after meal intake and lowering the blood glucose content significantly, inhibitors of dipeptidylpeptidase-IV prevent the degradation of these two peptides; hence, they enhance insulin secretion and lower blood glucose levels [160]. According to a 10-year prospective cohort study, the regular consumption of fermented fruits and vegetables can reduce the risk of diabetes due to the health benefits caused by fermented fruits and vegetables. In this regard, fermented fruits and vegetables include luteolin and isorhamnetin-3-O-glucoside, which are considered to be effective inhibitors of DPP-IV and α-glucosidase found in nature due to their phenolic acid compounds [161].

### 7.6. Comparison of Bioactivities Before and After Fermentation

Chemical changes and mechanisms caused by fermentation significantly alter the chemical composition of fruits and vegetables, leading to enhanced or novel bioactivities. These changes are primarily driven by microbial enzymatic activities, which breakdown complex macromolecules, synthesize new compounds, and modify existing ones. The resulting metabolites contribute to a range of health benefits, often through the involvement of distinct mechanisms compared to their unfermented counterparts. A comparison of such bioactivities is summarized in Table 3.

## 8. Food Security of Fermented Food

Fermentation technologies play an essential role in enhancing the food security of millions of poor, vulnerable people by improving food preservation, widening the range of raw materials that can be used to produce fermented food products, and reducing antimicrobial factors to levels that render foods safe for human consumption [162]. Moreover, many examples of fermentation by-products can be safely fed to animals as a nutritional supplement, further improving the livelihood systems [163]. Fermentation is a cheap and environmentally friendly technique of preserving perishable raw materials, which is accessible to even the most marginalized, landless, and physically disabled, rural and urban poor. For example, fruit and vegetables start to spoil immediately after harvesting, especially in the humid tropics, wherein the prevailing environmental conditions accelerate the decomposition process [164]. There are various options for preserving fresh fruit and vegetables, including drying, freezing, canning, and pickling, but many of these are inappropriate for small use cases. For example, small-scale canning of fruit and vegetables can have profound food safety implications given the potential for contamination with botulism, and freezing fruit and vegetables is not economically viable at a small scale [165]. However, the process of fermentation requires very little specialized equipment, either to perform or subsequently preserve the fermented food, and has had a big impact on nutritional habits, customs, and cultures [166]. As such, traditional fermentation still functions as an alternative to refrigeration or the safekeeping of food and is also directly used to make good of edible surpluses.

## 9. Safety Problems in Regard to Fermented Food and Current Solutions

Fermented foods have a good safety record, even in developing countries, where people prepare them without proper training in microbiology or chemistry in unsanitary, contaminated environments [167]. Millions of individuals consume them daily in both developed and developing countries. However, fermented foods do not solve other problems, such as those related to contaminated drinking water, heavily contaminated environments, improper personal hygiene, flies carrying disease organisms, unfermented foods likely to cause food poisoning or carrying human pathogens, and unfermented foods that are improperly cooked or stored [168]. Poorly fermented foods can be harmful. Implementing safety principles could improve the overall quality and nutritional value of the food supply, decrease dietary disorders, and increase resistance to intestinal and other diseases in newborns [134]. The fruit and vegetable industry must develop products considering quality and safety factors, while addressing consumer acceptance. Three key safety challenges occur during fermented fruit and vegetable production stages: biogenic amine, nitrite, and microbiological safety [169].

### 9.1. Biogenic Amines

Biogenic amines emerge as a result of the relationship between the pickles’ contents and the metabolic activity of bacteria. The biogenic amine content of fermented vegetables is directly connected to the biogenic amine content of the basic materials and the fermentation conditions [170]. Histamine, putrescine, and cadaverine are the principal biogenic amines present in fermented vegetables. Due to the difficulty of maintaining hygienic conditions during fermentation, homemade fermented vegetables may contain more biogenic amines than commercially produced ones. Reusing brine may also increase biogenic amine synthesis [171].

Bacteria strains capable of producing biogenic amines include *Enterococci*, *Lactobacilli*, *Streptococci*, *Pediococci*, and *Oenococci* [172]. Lactobacilli from some naturally fermented pickles can produce putrescine (PUT), cadaverine (CAD), and histamine (HIS). CAD and nitrite are related to Leuconostoc, while *Lactobacillus* and Pseudomonas are connected with tyramine (TYR). *Levilactobacillus brevis* primarily generates TYR [126]. It is feasible to lower the biogenic amine concentration of fermented vegetables by altering the fermentation conditions [173]. Although the salt content has some impact on the synthesis of biogenic amines in specific varieties of pickles, studies have indicated that adjusting the salt concentration and temperature has a limited impact on preventing the formation of biogenic amines in pickles. Changing the composition of pickles with a relatively low precursor of biogenic amines could reduce their final content [174].

### 9.2. Nitrite

Nitrite, another dangerous chemical in fermented fruit and vegetables, can produce cancer-causing nitrosamines and cause cancer of the digestive tract. The GB 2762-2017 [175] standards (Food Safety National Standard Food Pollutant Limits of China) set a maximum nitrite concentration of 4 mg/kg for raw vegetables and 20 mg/kg for fermented vegetables [176]. The level of nitrite in various pickles changes depending on the components used. According to recent research, the most significant nitrite level has been discovered in fermented cabbage, followed by fermented mustard, bamboo, and radishes [177]. The nitrite concentration is connected to the composition of the microbial population in pickles. Genome sequencing has revealed a direct correlation between the *Lactobacillus* spp. relative abundance and nitrite levels, suggesting that modulating the microbial community composition (e.g., through the addition of seasoning) could effectively reduce the nitrite concentration in fermented vegetables [178].

Due to the formation of lactic acid bacteria in fermented fruit and vegetables during fermentation, nitrite can be destroyed spontaneously. Various strains that breakdown nitrite have been identified in different pickles [179]. These bacterial strains positively affect nitrite degradation when used as a starter culture for producing pickles and contribute to the excellent qualities of fermented fruit and vegetables, including beneficial qualities due to the presence of *Lacticaseibacillus casei* subsp. *rhamnosus* LCR 6013, *Lactiplantibacillus plantarum* ZJ316, *Stachys sieboldii* Miq., and *Lactobacillus coryniformis* [180]. Generally, nitrite breakdown can occur below a pH of 4. A low pH value arises from the accumulation of organic acid produced by lactic acid bacteria, including lactic acid, acetic acid, butyric acid, tartaric acid, succinic acid, citric acid, and malic acid [57]. The nitrite content in mixed-strain fermented pickles is lower than in single-strain fermented ones because of the formation of organic acids. In addition, the accumulation of organic components can increase the flavor of pickles. Hence, naturally fermented or mixed-strain fermented pickles have outstanding flavors [181].

### 9.3. Microbial Safety

During the manufacturing of fermented foods, primarily pickles, there are possible safety problems that can occur related to microbial variables that could affect customers (Table 4). It is necessary to control the presence of pathogens at two critical points [182]. The first point is during the pre-treatment procedure, which concerns the need to reduce the presence of miscellaneous bacteria, especially those that have strong biofilm-forming abilities; if these microorganisms are propagated and form a biofilm during the pickling procedure, the flavor of the fermented fruit and vegetables would be affected [183]. The other key point is at the end of the fermentation phase. One potential action in this regard is to terminate the fermentation process, to avoid the adverse effect on the flavor and texture of the fermented fruit and vegetables caused by excessive fermentation, but the most critical purpose of such action is to extend the shelf life of the fermented fruit and vegetables and to maintain their safety for consumption by consumers; fermented fruit and vegetables have always been considered to be ready-to-eat products in the market; for most homemade fermented fruit and vegetable products, the sterilization process is always missing [184]. The safety of fermented foods can be enhanced through the use of proper fermentation techniques, which minimize the risk of pathogenic contamination and ensure that a high-quality product is created.

Heat treatment is widely utilized in regard to the industrial sterilization of fermented fruit and vegetables at both critical points in the process. However, industrial sterilization affects the volatile components and texture of pickles, thereby diminishing consumer acceptance [185]. To solve these concerns, investigating novel non-thermal sterilizing technologies to control hazardous germs has generated significant attention. In the meantime, in regard to conventional fermentation, a high sodium concentration is commonly used for microbial control; the penetration of salt into fermented substrates depends on various factors, and a long period of salting could increase the risk of microbial deterioration. Some non-thermal technologies can also accelerate the salting procedure [186]. The ongoing research into fermentation techniques continues to uncover innovative methods to enhance food safety, while preserving the sensory qualities of fermented products.

**Table 4 foods-14-02155-t004:** Microbial safety during the fermentation process, safety concerns, control points, traditional methods, and emerging non-thermal technologies.

Critical Control Points	Safety Concerns	Traditional Control Methods	Emerging Non-Thermal Technologies	Refs.
Pre-treatment Procedure	Biofilm-forming bacteria may affect flavor, texture, and safety of fermented products	Heat treatment to reduce initial microbial load	High-pressure processing (HPP), ozone treatment, pulsed electric fields (PEFs)	[174]
Salting Process	High salt levels control microbes, but may affect health and increase microbial deterioration risk	High sodium concentration and long salting duration	PEFs, ultrasound to accelerate salt penetration and reduce salt usage	[187]
Packaging and Storage	Pathogens in ready-to-eat fermented foods can pose health risks without final sterilization	Pasteurization before packaging	Pulsed light treatment, ozone treatment, HPP	[188]
Termination of Fermentation	Over-fermentation can negatively impact flavor and texture; pathogens can reduce shelf life	Heat treatment to stop fermentation and extend shelf life	Cold plasma, UV-C irradiation, ultrasound treatment	[189]

## 10. Thoughts on the Future of Fruit and Vegetable Fermentation

The future of fruit and vegetable fermentation is on the verge of a significant transformation by incorporating cutting-edge technologies and sustainable methodologies. Intelligent fermentation technologies, which utilize artificial intelligence (AI) and automation, facilitate the real-time surveillance and regulation of fermentation variables, including the pH, temperature, and nutrient concentrations, guaranteeing uniform product quality and the improved retention of bioactive compounds [190]. Precision microbiome engineering offers the potential to develop customized microbial strains or consortia designed to deliver specific health benefits, such as improved gut health, immunity, or mental wellness, thereby aligning fermented products with personalized nutrition. Additionally, exploring the fermentation of lesser-known and underutilized fruits and vegetables enhances dietary diversity, supports biodiversity, and strengthens local food systems. Fermentation adds value to such produce, reduces post-harvest losses, and empowers small-scale farmers [191]. Furthermore, zero-waste fermentation models are gaining importance, wherein by-products like peels and pomace are repurposed into value-added ingredients, such as dietary fiber and natural preservatives, contributing to circular economy goals. Collectively, these advancements promise to transform traditional fermentation into a highly innovative, health-driven, and environmentally conscious food processing strategy.

## 11. Conclusions

Fermented fruits and vegetables represent a highly promising pathway for the advancement of functional foods, providing augmented nutritional benefits, an extended shelf life, and advantageous probiotic characteristics. As consumer demand for health-promoting, minimally processed food products continues to increase, the prospects for these commodities within the global market are substantial. Microbial fermentation processes, which include lactic acid bacteria and various probiotic microorganisms, play a critical role in augmenting the bioavailability of nutrients and the provision of health benefits, such as enhancements in gut health, antimicrobial efficacy, and anti-inflammatory reactions. Future investigations should concentrate on refining fermentation methodologies, examining novel fermentation strains, and elucidating the interactions of the microbiome within fermented plant-based food matrices. In the long term, fermented fruits and vegetables might evolve into vital constituents of sustainable, health-oriented dietary patterns on a global scale.

## Figures and Tables

**Figure 1 foods-14-02155-f001:**
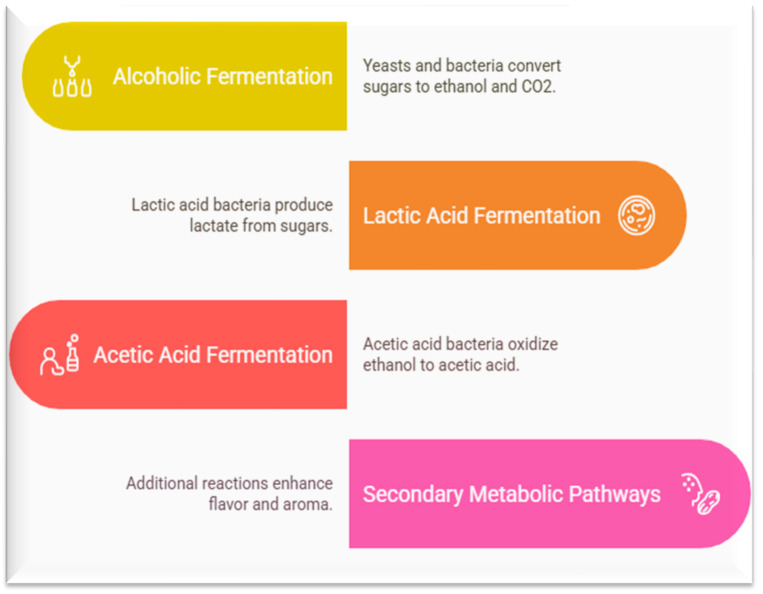
Metabolic processes occurring in fruit and vegetable fermentation

**Figure 2 foods-14-02155-f002:**
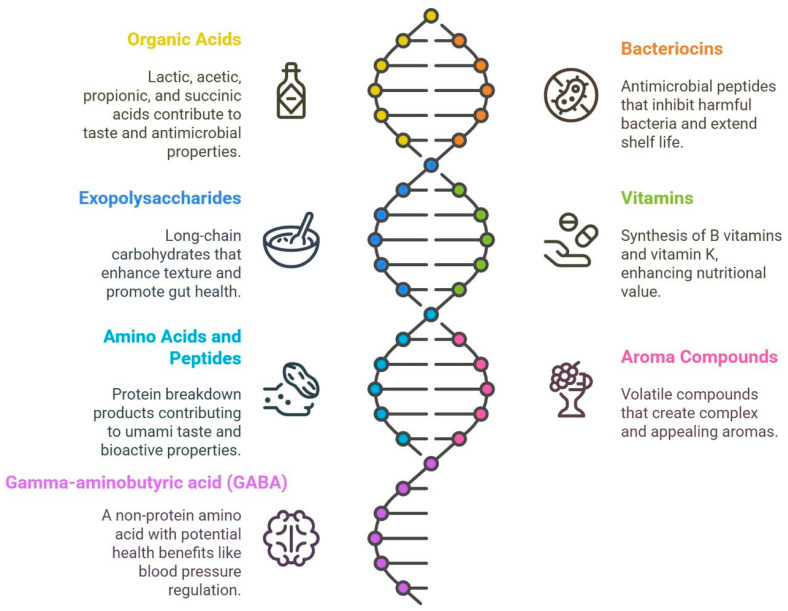
Key metabolites produced during lactic acid fermentation.

**Figure 3 foods-14-02155-f003:**
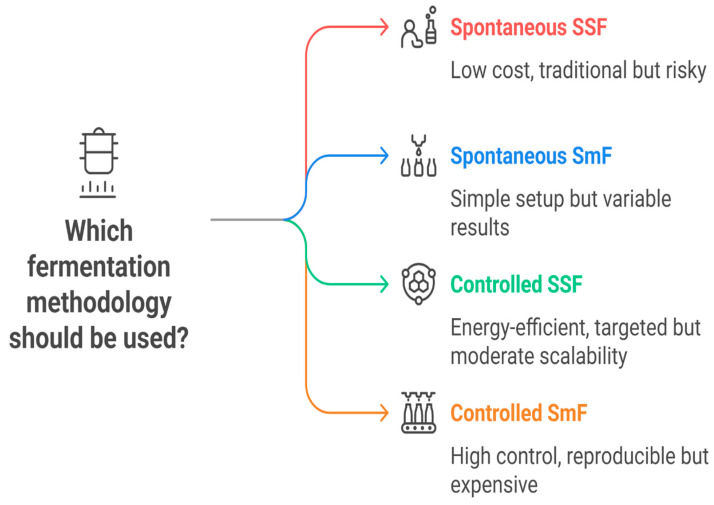
Classification and characteristics of fermentation techniques used in the fermentation of fruits and vegetables.

**Figure 5 foods-14-02155-f005:**
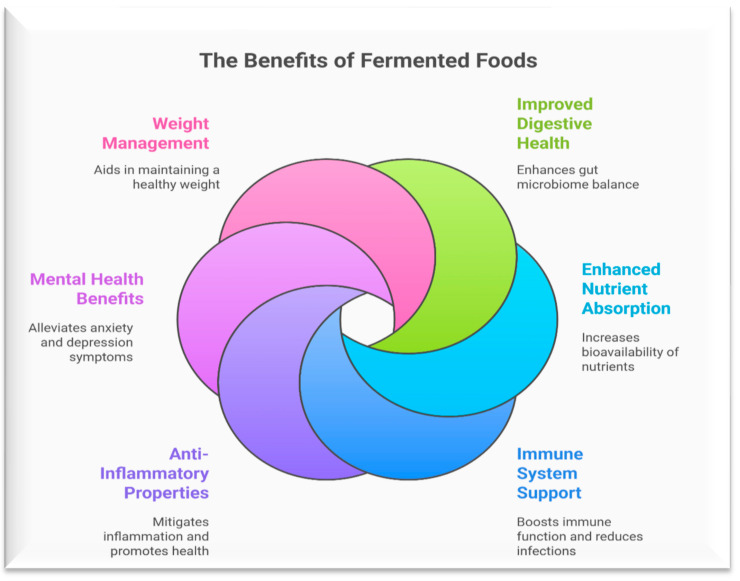
Several health benefits of fermented foods in the human diet.

**Table 1 foods-14-02155-t001:** Various fermented products made from fruits and vegetables.

Fermented Product	Fruits and Vegetables Used	Lactic Acid Bacteria	Type of Fermentation	Refs.
Sauerkraut	Cabbage	*Leuconostoc mesenteroides*, *Levilactobacillus brevis*	Lactic Acid Fermentation	[54]
Fermented Cucumbers	Cucumbers	*Pediococcus pentosaceus*, *Lactiplantibacillus plantarum*	Lactic Acid Fermentation	[55]
Fermented Caper Berries	Capers	*Lactiplantibacillus pentosus*, *Limosilactobacillus fermentum*, *Pediococcus pentosaceus*, *Lactobacillus paraplantarum*, *Enterococcus faecium*	Lactic Acid Fermentation	[56]
Kimchi	Cabbage, Radish	*Lactobacillus kimchiensis*, *Leuconostoc citreum*, *Lacticaseibacillus gasicomitatum*, *Lactobacillus bavaricus*, *Weissella confusa*, *Weissella kimchii*, *Latilactobacillus curvatus*, *Lactobacillus sakei*, *Loigolactobacillus maltaromicus*	Mixed Fermentation	[57]
Gundruk	Cabbage, Mustard Leaves, Cauliflower Leaves	*Lacticaseibacillus casei*, *Leuconostoc pseudoplantarum*, *Limosilactobacillus fermentum*, *Pediococcus pentosaceus*	Lactic Acid Fermentation	[58]
Sinki	Radish Roots	*Limosilactobacillus fermentum*, *Lactiplantibacillus plantarum*, *Levilactobacillus brevis*, *Levilactobacillus fallax*	Lactic Acid Fermentation	[59]
Khalpi	Cucumber	*Lactiplantibacillus plantarum*, *Pediococcus* sp., *Levilactobacillus brevis*, *Levilactobacillus fallax*	Lactic Acid Fermentation	[60]
Kanji	Carrot and Beetroot	*Levilactobacillus brevis*	Lactic Acid Fermentation	[61]
Wine	Grape	*Saccharomyces cerevisiae*	Alcoholic Fermentation	[62]
Apple Cider	Apple	*Enterococcus* spp.	Alcoholic and Acetic Fermentation	[63]
Fermented Strawberry Juice	Strawberry	*Lactiplantibacillus plantarum*, *Lactobacillus acidophilus*	Lactic Acid Fermentation	[64]
Fermented Blueberry Juice	Blueberry	*Lacticaseibacillus rhamnosus*, *Lactiplantibacillus plantarum*	Lactic Acid Fermentation	[65]

**Table 3 foods-14-02155-t003:** A comparison of bioactivities before and after fermentation.

Comparison Item	Before Fermentation	After Fermentation	Hypothetical Mechanisms
Enhanced bioavailability and bioactivity of phenolic compounds	Phenolic compounds are present, but their absorption and utilization by the human body can be limited due to their bound forms and complex matrix interactions	The hydrolysis of glycosides and esters increases the concentration of free, more readily absorbable phenolic aglycones. This often leads to a significant increase in antioxidant capacity, as demonstrated by higher DPPH radical scavenging activity and ferric-reducing antioxidant power (FRAP) in fermented products compared to the raw materials	The enhanced antioxidant activity is attributed to the increased concentration of free phenolic compounds, which can directly scavenge free radicals, chelate metal ions, and upregulate endogenous antioxidant enzymes. The smaller molecular size of aglycones facilitates their absorption across the intestinal barrier, allowing them to exert systemic antioxidant and anti-inflammatory effects
Production of bioactive peptides and free amino acids	Proteins are present, but their bioactive potential is largely latent within the larger protein structure	Fermented products often exhibit enhanced antihypertensive (ACE inhibitory), antioxidant, and immunomodulatory activities, due to the release of specific bioactive peptides. For example, peptides with ACE-inhibitory activity have been identified in fermented vegetable products, contributing to blood pressure regulation	Bioactive peptides can exert their effects by binding to specific receptors, modulating enzyme activities, or acting as signaling molecules. For instance, ACE-inhibitory peptides compete with angiotensin I in regard to binding to the angiotensin-converting enzyme, thereby preventing the formation of angiotensin II, a potent vasoconstrictor
Synthesis of vitamins	Vitamin content is dependent on the raw material and can vary	Fermented products often show increased levels of folate, riboflavin, and vitamin B12, which are essential for various metabolic processes, including DNA synthesis, energy production, and nerve function. This enhancement directly contributes to the improved nutritional status of such foods	The increased vitamin content directly contributes to their respective physiological roles. For example, increased folate levels support cell division and growth, while higher vitamin B12 contributes to red blood cell formation and neurological function
Production of short-chain fatty acids (SCFAs)	Raw fruits and vegetables contain dietary fibers that are precursors to SCFAs upon gut microbial fermentation	Fermented products can contain pre-formed SCFAs or promote their production in the gut. Butyrate, in particular, is a crucial energy source for colonocytes and plays a significant role in maintaining gut barrier integrity and modulating immune responses	SCFAs, particularly butyrate, exert their beneficial effects by acting as signaling molecules, modulating gene expression, and influencing immune cell differentiation. They contribute to improved gut health by strengthening the intestinal barrier, reducing inflammation, and promoting the growth of beneficial gut bacteria
Reduction of anti-nutritional factors	Anti-nutritional factors can chelate essential minerals (e.g., iron, zinc, calcium), reducing their absorption	The reduction in anti-nutritional factors leads to the increased bioavailability of minerals, enhancing the nutritional value of the fermented product	By degrading anti-nutritional factors, fermentation liberates bound minerals, making them more accessible for absorption in the gastrointestinal tract. This directly improves the nutritional impact of consuming these foods

## Data Availability

No new data were created or analyzed in this study. Data sharing is not applicable to this article.

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
