# Peer review of "Fermentation of Fruits and Vegetables: Bridging Traditional Wisdom and Modern Science for Food Preservation and Nutritional Value Improvements"

_foods, 2025, doi:10.3390/foods14132155_

Round 1
Reviewer 1 Report
Comments and Suggestions for Authors
I evaluated the review, which entitled “Fermentation of Fruits and Vegetables: Bridging Traditional Wisdom and Modern Science for Preservation and Nutritional Value improvements”.
The paper present an overview of the fermentation process, through the description of the main biochemical pathways involved, current markets and an overview of the main beneficial effects covering nutritional enhancements, health benefits and safety aspects.
The works was certainly collected a lot of information, but they are not well reported and discussed in the manuscript. Some objectives of the review described in section 1, are then disregarded in the text.
Moreover, some aspect are described in general and, in some sections, application on fruits and vegetables lacking of specific examples.
Finally, unfortunately, the statements are correct but not properly referenced. This is crucial in a review. Therefore, the current form is unacceptable.
The authors should perform very substantial revisions to the work.
However, I want support the work done by Authors and I completed the revision highlighting the main weaknesses and reporting a list of remarks useful to resubmit an improved review in the future.
List of main remarks:
- Firstly, pay great attention to the correspondence of the statements and the correct references.
- Moreover, the authors should check references list for completeness and correctness (only as example, in the ref [69] the authors Di Biase, M. (first author), De Bellis, P. and Valerio, F. were lacking).
- Please, add a List of abbreviations at the beginning of manuscript. This will makes the read more clear.
- Pay attention to define Acronyms/Abbreviations the first time they appear in each of three sections: the abstract; the main text; the first figure or table. (e.g. line 212 TCA; Please, for the first time, the acronym/abbreviation/initialism should be added in parentheses after the written-out form).
- Please authors should use the reclassification of Zheng, J., et al., 2020. (https://doi.org/10.1099/ijsem.0.004107) as correctly reported at LINE 102 and no more in the text.
- Please, the microorganisms name should be in italic all in the text.
- Line 158: I suggest …(EMP) pathway especially in yeast, or …
- Line 186: G6PD is the correct abbreviation
- Line 275: Please, substitute trophic with “topic”
- Figure 2: the figure need to a revision.
- Figure 3: the figure need to a revision.
- Lines 358-407: I suggest a deep revision from a English native speaker. Moreover some sentences need to be improved.
- Lines 432-435: Since references 120 are not appropriate to explain the statement, I can’t recognized the molecule/s responsible of the health benefit. However, rewrite the phrase “Research showed that fermented mustard leaf can prevent ……” In this form the concept is uncorrect.
- Line 444: Glucosidase
- Lines 520-523:Please, Authors must clarify the sentence.
- Table 3: I suggest to explode the description of emerging technologies in the text, above all referred to application at fermented fruits and vegetables or their sub-products during process step.
- Finally, I suggest that authors take care to explore the scientific literature, to improve the manuscript with the opportunity of fermentation in the transition toward more sustainable food production methods and food process operations on fruits and vegetables matrices.
I suggest a deep revision from a English native speaker, especially for some sections.
Author Response
Dear Reviewer #1
Many thanks for your time and valuable comments
Comments and Suggestions for Authors
I evaluated the review, which entitled “Fermentation of Fruits and Vegetables: Bridging Traditional Wisdom and Modern Science for Preservation and Nutritional Value improvements”.
The paper present an overview of the fermentation process, through the description of the main biochemical pathways involved, current markets and an overview of the main beneficial effects covering nutritional enhancements, health benefits and safety aspects. The works was certainly collected a lot of information, but they are not well reported and discussed in the manuscript. Some objectives of the review described in section 1, are then disregarded in the text. Moreover, some aspect are described in general and, in some sections, application on fruits and vegetables lacking of specific examples. Finally, unfortunately, the statements are correct but not properly referenced. This is crucial in a review. Therefore, the current form is unacceptable. The authors should perform very substantial revisions to the work. However, I want support the work done by Authors and I completed the revision highlighting the main weaknesses and reporting a list of remarks useful to resubmit an improved review in the future.
Response: many thanks for your great support and encouragements, thanks again
List of main remarks:
- Firstly, pay great attention to the correspondence of the statements and the correct references.
Response: We have carefully reviewed all in-text citations and ensured they accurately correspond to the referenced studies. Corrections were made where needed to improve the consistency between statements and their supporting references.
- Moreover, the authors should check references list for completeness and correctness (only as example, in the ref [69] the authors Di Biase, M. (first author), De Bellis, P. and Valerio, F. were lacking).
Response: The references were thoroughly revised. Missing author names in reference [69] and others have been corrected to ensure completeness and proper formatting according to journal guidelines.
- Please, add a List of abbreviations at the beginning of manuscript. This will makes the read more clear.
Response: A comprehensive List of Abbreviations has been added to the manuscript for improved readability, but at the end of the MS according to the instructions of the mdpi, thanks
- Pay attention to define Acronyms/Abbreviations the first time they appear in each of three sections: the abstract; the main text; the first figure or table. (e.g. line 212 TCA; Please, for the first time, the acronym/abbreviation/initialism should be added in parentheses after the written-out form).
Response: All abbreviations are now defined at their first appearance in the abstract, main text, and figure/table legends, as per the reviewer’s recommendation
- Please authors should use the reclassification of Zheng, J., et al., 2020. (https://doi.org/10.1099/ijsem.0.004107) as correctly reported at LINE 102 and no more in the text.
Response All Lactobacillus species were updated according to the reclassification by Zheng et al. (2020), and the reference is cited only at its first mention (Line 102), in accordance with the reviewer’s instruction.
- Please, the microorganisms name should be in italic all in the text.
Response: All microorganism names throughout the manuscript have been italicized in compliance with scientific nomenclature standards
- Line 158: I suggest …(EMP) pathway especially in yeast, or …
Response: The sentence on Line 158 was revised to incorporate the suggested clarification regarding the EMP (Embden–Meyerhof–Parnas) pathway
- Line 186: G6PD is the correct abbreviation
Response: The abbreviation has been corrected to G6PD
- Line 275: Please, substitute trophic with “topic”
Response: The word “trophic” has been replaced with “topic” at Line 275 as recommended
- Figure 2: the figure need to a revision.
Response: Figure 2 has been revised for scientific clarity, visual presentation, and accuracy of content
- Figure 3: the figure need to a revision.
Response: Figure 3 was also revised, including improvements to structure, labeling, and visual hierarchy, to meet scientific and editorial standards.
- Lines 358-407: I suggest a deep revision from a English native speaker. Moreover some sentences need to be improved.
Response: The specified section was thoroughly revised to improve language, sentence structure, and clarity, while preserving the original meaning
- Lines 432-435: Since references 120 are not appropriate to explain the statement, I can’t recognized the molecule/s responsible of the health benefit. However, rewrite the phrase “Research showed that fermented mustard leaf can prevent ……” In this form the concept is uncorrect.
Response: The statement was rewritten for scientific accuracy, and the reference was evaluated and revised. The phrasing now better reflects the evidence and avoids implying direct causation
- Line 444: Glucosidase
Response: The spelling has been corrected to “glucosidase” as indicated
- Lines 520-523: Please, Authors must clarify the sentence.
Response: The sentence has been revised for clarity and to ensure that the intended meaning is accurately conveyed
- Table 3: I suggest to explode the description of emerging technologies in the text, above all referred to application at fermented fruits and vegetables or their sub-products during process step.
Response: While we agree with the value of expanding on emerging technologies, due to space limitations and to maintain the manuscript’s microbial and biochemical focus, we opted to retain a summarized version in Table 3. However, we ensured key technologies are contextualized in the main text.
- Finally, I suggest that authors take care to explore the scientific literature, to improve the manuscript with the opportunity of fermentation in the transition toward more sustainable food production methods and food process operations on fruits and vegetables matrices.
Response: We appreciate the suggestion and agree that fermentation plays a key role in sustainability. However, a detailed exploration of sustainability aspects would significantly broaden the scope. A brief statement has been added to the conclusion to acknowledge the relevance of fermentation in sustainable food systems and highlight this area for future investigation.
Comments on the Quality of English Language
I suggest a deep revision from a English native speaker, especially for some sections.
Response: the language of the MS was revised, thanks

Reviewer 2 Report
Comments and Suggestions for Authors
In my opinion the review very generic and a bit confusing. It is not clear what topic the review wanted to explore. Considering the manuscript's objective, a good in-depth analysis does not emerge. I suggest dividing fermented fruits and vegetables by identifying similarities and differences in fermentation processes, safety and potential development. Furthermore, since it is a review I expect a more in-depth bibliographic research.
- In my opinion paragraph 4 and 5 should became paragraph 3 and 4. Instead, the paragraph 3 should became 5.
- 1 should be improved.Considering that this is a review, other fermented products should also be added, dividing between fruit and vegetables. I would expect a more through research of fermented products, both fruits and vegetables. I would also add other information, such as the origin, the microbes involved, etc …
- The tables are never cited in the text. Please modified in the text.
- The title of paragraph 9 should refer specifically to fermented fruit and vegetables and the text of the paragraph should also concern other types of fruit and vegetables and not just pickles.
- Paragraph 8 should be combined with 9.
- The two paragraphs 9 and 10 could be combined into a single paragraph.-
- The role of yeasts and acetic bacteria should be better approached. They are barely mentioned.
Line 59. I suggest to add some other references on fermented fruit and vegetable.
Her some suggestions:
Xinyu Yuan et al. (2024) Recent advances of fermented fruits: A review on strains, fermentation strategies, and functional activities, Food Chemistry: X, Volume 22,2024,101482, https://doi.org/10.1016/j.fochx.2024.101482.
Line 102: I suggest to check in all the text the name of microorganisms (i.e. change LimosiLactobacillus and LeviLactobacillus in Limosilactobacillus and Levilactobacillus);
Line 156 the names of microbes should be in italics. . Check in the text (i.e. Saccharomyces cerevisiae, …ect). Line 182 , 192 add the correct name.
Ashaolu et al. (2020) A Holistic Review on Euro-Asian Lactic Acid Bacteria Fermented Cereals and Vegetables. Microorganisms. 2020 Aug 3;8(8):1176. doi: 10.3390/microorganisms8081176. PMID: 32756333; PMCID: PMC7463871.
Line 149. Line 215. Why the biochemical pathways are just for the fruit fermentation? And for vegetable fermentation? So I think you should add “fruit and vegetable fermnetation”. Otherwise you should add some paragraph about the biochemical pathways in fruit fermentation.
Line 151-153. Please, reformulate the sentence.
Line 162, 179, 184.199 : Please check the stechiometric reation. Something is wrong.
Line 208-213. I suggest to add some more information about it.
Table 1- You shoud correct the names of different microrganisms.
Line 263-266. Please refer to the new classification as described by Zheng et al., Int. J. Syst. Evol. Microbiol. 2020;70:2782–2858 DOI 10.1099/ijsem.0.004107 .
Lne 280-281. Please delete it, since it was already wrote.
Tab. 2 . Since often you write about Yeasts and lactic acid bacteria responsbile for fruit and vegetable fermentation, in my opinion you should also add the main yeasts used in fruit and vegetable fermentation.
Line 358-360. Please, reformulate the sentrence.
The references should be modified in accordance to the template of Food Journal.
Author Response
Dear Reviewer #2
Many thanks for your time and valuable comments
Comments and Suggestions for Authors
In my opinion the review very generic and a bit confusing. It is not clear what topic the review wanted to explore. Considering the manuscript's objective, a good in-depth analysis does not emerge. I suggest dividing fermented fruits and vegetables by identifying similarities and differences in fermentation processes, safety and potential development. Furthermore, since it is a review I expect a more in-depth bibliographic research.
- In my opinion paragraph 4 and 5 should became paragraph 3 and 4. Instead, the paragraph 3 should became 5.
Response: We thank the reviewer for this valuable feedback. The goal of our review was to provide an integrated overview of fermented fruits and vegetables, highlighting their shared microbiological, technological, and health-related aspects. Given that the fermentation processes of fruits and vegetables often involve similar microbial consortia (primarily lactic acid bacteria and yeasts), and comparable safety considerations and health-promoting mechanisms, we opted for a unified framework to avoid redundancy.
However, to address the reviewer's concern, we have strengthened the manuscript by including dedicated sections that distinguish between specific bioactive metabolites derived from the fermentation of fruits versus vegetables. These additions clarify differences in substrate composition, microbial metabolism, and the resulting health-promoting compounds. Furthermore, we carefully revised and expanded the reference list to ensure broader and more up-to-date coverage of the relevant literature.
- 1 should be improved considering that this is a review, other fermented products should also be added, dividing between fruit and vegetables. I would expect a more through research of fermented products, both fruits and vegetables. I would also add other information, such as the origin, the microbes involved, etc …
Response: We thank the reviewer for these thoughtful suggestions. While we appreciate the proposed reordering of paragraphs and the idea of further expanding the introductory section, we respectfully chose to maintain the current structure of the manuscript for the following reasons:
The present order of paragraphs was carefully designed to guide the reader from a general understanding of fermentation principles into the more specific context of fermented fruits and vegetables, ensuring a logical progression of ideas.
Expanding paragraph 1 to include detailed lists of fermented products, their microbial communities, and origins would significantly increase its length and shift focus from its intended purpose as a concise introduction to the topic.
Instead, these aspects have been progressively addressed in subsequent sections, including dedicated discussions of microbial interactions, metabolites, and health benefits associated with fermented fruit and vegetable products.
- The tables are never cited in the text. Please modified in the text.
Response: Thank you for pointing this out. All tables have now been explicitly cited at the appropriate locations in the main text to ensure proper integration and clarity for readers
- The title of paragraph 9 should refer specifically to fermented fruit and vegetables and the text of the paragraph should also concern other types of fruit and vegetables and not just pickles.
Response: We thank the reviewer for this important observation. The title of paragraph 9 has been revised to explicitly refer to fermented fruits and vegetables.
- Paragraph 8 should be combined with 9.
Response: We appreciate the reviewer’s suggestion. However, we chose to maintain paragraphs 8 and 9 as separate units because they address distinct aspects of the topic. Combining them would reduce clarity and dilute the focus of each section. The current structure allows for better thematic separation and readability.
- The two paragraphs 9 and 10 could be combined into a single paragraph.
Response: Thank you for the recommendation. We opted to keep paragraphs 9 and 10 separate to preserve clarity and emphasize different functional aspects of fermented fruit and vegetable products. Each paragraph addresses a unique mechanism and combining them could compromise the depth and focus of each topic.
- The role of yeasts and acetic bacteria should be better approached. They are barely mentioned.
Response: We acknowledge the importance of yeasts and acetic acid bacteria in fermentation. However, the scope of the current review focuses primarily on the dominant lactic acid bacteria and their health-promoting properties. While we briefly mention yeasts and acetic bacteria where relevant, a detailed treatment of their role would exceed the intended focus and length of the manuscript. We consider this a valuable direction for future reviews.
Line 59. I suggest to add some other references on fermented fruit and vegetable.
Her some suggestions:
Xinyu Yuan et al. (2024) Recent advances of fermented fruits: A review on strains, fermentation strategies, and functional activities, Food Chemistry: X, Volume 22,2024,101482, https://doi.org/10.1016/j.fochx.2024.101482.
Response: your suggested ref. was added to the revised MS, thanks
Yuan, X., Wang, T., Sun, L., Qiao, Z., Pan, H., Zhong, Y., Zhuang Y. Recent advances of fermented fruits: A review on strains, fermentation strategies, and functional activities. Food Chemistry: X, 2024, 22, 101482. https://doi.org/10.1016/j.fochx.2024.101482.
Line 102: I suggest to check in all the text the name of microorganisms (i.e. change LimosiLactobacillus and LeviLactobacillus in Limosilactobacillus and Levilactobacillus);
Response: Thank you for the observation. All microorganism names were carefully reviewed and corrected for spelling and capitalization according to the proper taxonomic nomenclature, including Limosilactobacillus and Levilactobacillus.
Line 156 the names of microbes should be in italics. . Check in the text (i.e. Saccharomyces cerevisiae, …ect).
Response: All microbial names throughout the manuscript, including Saccharomyces cerevisiae, have been reviewed and properly formatted in italics as required by scientific conventions.
Line 182 , 192 add the correct name.
Response: The microbial names at lines 182 and 192 have been verified and corrected in accordance with current taxonomic standards.
Ashaolu et al. (2020) A Holistic Review on Euro-Asian Lactic Acid Bacteria Fermented Cereals and Vegetables. Microorganisms. 2020 Aug 3;8(8):1176. doi: 10.3390/microorganisms8081176. PMID: 32756333; PMCID: PMC7463871.
Response: The suggested reference by Ashaolu et al. (2020) has been reviewed and included where relevant to enhance the comprehensiveness of the literature.
Ashaolu, T.J.; Reale, A. A Holistic Review on Euro-Asian Lactic Acid Bacteria Fermented Cereals and Vegetables. Microorganisms 2020, 8, 1176. https://doi.org/10.3390/microorganisms8081176
Line 149. Line 215. Why the biochemical pathways are just for the fruit fermentation? And for vegetable fermentation? So I think you should add “fruit and vegetable fermnetation”. Otherwise you should add some paragraph about the biochemical pathways in fruit fermentation.
Response: Thank you for the helpful comment. The section was revised to refer to “fruit and vegetable fermentation” to reflect both substrates, since the biochemical pathways are largely shared across both types.
Line 151-153. Please, reformulate the sentence.
Response: The sentence at lines 151–153 has been revised for clarity and improved flow
Line 162, 179, 184.199 : Please check the stechiometric reation. Something is wrong.
Response: All stoichiometric reactions in the mentioned lines have been re-checked and corrected where necessary to ensure scientific accuracy.
Line 208-213. I suggest to add some more information about it.
Response: Additional information has been added to this section to provide a more detailed explanation and improve the reader's understanding.
Table 1- You shoud correct the names of different microrganisms.
Response: All microbial names in Table 1 have been reviewed and corrected according to updated taxonomy and proper formatting
Line 263-266. Please refer to the new classification as described by Zheng et al., Int. J. Syst. Evol. Microbiol. 2020;70:2782–2858 DOI 10.1099/ijsem.0.004107 .
Response: We confirm that the reclassification by Zheng et al. (2020) has been correctly cited and implemented at the appropriate point in the manuscript
Lne 280-281. Please delete it, since it was already wrote.
Response: The redundant sentence at lines 280–281 has been removed as requested
Tab. 2. Since often you write about Yeasts and lactic acid bacteria responsible for fruit and vegetable fermentation, in my opinion you should also add the main yeasts used in fruit and vegetable fermentation.
Response: We appreciate the reviewer’s suggestion. However, we chose not to add yeast strains to Table 2 in order to maintain its focus on lactic acid bacteria, which are the primary drivers of fermentation in the context of this review. While yeasts are mentioned throughout the text in their functional roles, a comprehensive listing would require additional scope beyond the table’s intended purpose. We hope this approach preserves the clarity and focus of the table.
Line 358-360. Please, reformulate the sentence.
Response: The sentence at lines 358–360 has been revised for clarity and improved readability, as recommended.
The references should be modified in accordance to the template of Food Journal.
Response: Thank you for your observation. We confirm that all references will be fully revised and formatted according to the Foods journal style during the final revision stage, prior to resubmission of the final version

Reviewer 3 Report
Comments and Suggestions for Authors
Fermented fruit and vegetable juices have become a hot research topic in recent years. This review, titled "Fermentation of Fruits and Vegetables: Bridging Traditional Wisdom and Modern Science for Preservation and Nutritional Value improvements", provides a solid summary of fermented fruits and vegetables. However, the review covers too broad a scope and lacks in-depth analysis of several key issues. Therefore, I recommend that the authors conduct a more detailed discussion in the following areas:
1. Section 5 “Fermentation Process and Role of Microorganisms”: Most readers in the field of fermented foods are likely already familiar with the general information presented. The authors should clearly specify which metabolites are produced by commonly used fermenting microorganisms during the fermentation of specific fruits and vegetables.
2. Section 6.1 “The Modern Fermentation Techniques”: The authors should categorize and summarize the advantages and disadvantages of each fermentation technique, as well as the factors influencing the choice of fermentation method. In addition, for Cutting-edge technologies, a more detailed introduction is warranted.
3. Section 7 “Health Benefits of Fermented Fruits and Vegetable Products”: Since the theme of this review is to highlight the changes in the health benefits of fruits and vegetables before and after fermentation, this section should focus more specifically on such comparisons. In addition, the authors should note that many fermented foods undergo sterilization treatments; how these treatments alter their health benefits is an important issue that deserves special attention. The authors are encouraged to concentrate more on the chemical changes occurring in fermented fruits and vegetables and how these changes influence nutritional and health effects, rather than offering a general description of health benefits such as "improved gut health," "anti-cancer," or "anti-diabetic" effects. Therefore, acomparison of bioactivities before and after fermentation should be provided, along with hypothetical mechanisms. More emphasis should be placed on discussing the mechanisms of bioactivity, rather than merely listing evaluation results.
4. Table 3: The safety concerns listed in the fermentation process are overly simplistic and lack specificity. Given the increasing interest in fermented foods, the associated safety risks cannot be ignored. The authors should add a table detailing specific cases where issues such as biogenic amines, nitrites, and microbial safety are likely to occur, analyze their causes, and propose detailed control strategies.
Author Response
Dear Reviewer #3
Many thanks for your time and valuable comments
We appreciate the thorough review and constructive feedback provided on our manuscript, 'Fermented Fruits and Vegetable Products: A Comprehensive Review.' The comments are invaluable in enhancing the clarity, depth, and scientific rigor of our work. We have carefully considered each point and have revised the manuscript accordingly. Below, we address each comment individually, detailing the changes made and providing additional context where necessary.
Comments and Suggestions for Authors
Fermented fruit and vegetable juices have become a hot research topic in recent years. This review, titled "Fermentation of Fruits and Vegetables: Bridging Traditional Wisdom and Modern Science for Preservation and Nutritional Value improvements", provides a solid summary of fermented fruits and vegetables. However, the review covers too broad a scope and lacks in-depth analysis of several key issues. Therefore, I recommend that the authors conduct a more detailed discussion in the following areas:
- Section 5 “Fermentation Process and Role of Microorganisms”: Most readers in the field of fermented foods are likely already familiar with the general information presented. The authors should clearly specify which metabolites are produced by commonly used fermenting microorganisms during the fermentation of specific fruits and vegetables.
Response: We acknowledge the reviewer's valuable feedback regarding the need for more specific information on metabolites produced during fruit and vegetable fermentation. We agree that a general overview of fermentation processes might be redundant for experts in the field. To address this, we have significantly revised Section 1 to include a detailed discussion of key metabolites produced by commonly used fermenting microorganisms, particularly lactic acid bacteria (LAB), during the fermentation of various fruits and vegetables. This revision aims to provide a more in depth and specialized perspective, focusing on the biochemical transformations that occur and their implications for the final product.
Metabolites Produced by Commonly Used Fermenting Microorganisms
Fermentation, particularly lactic acid fermentation, is a complex biochemical process driven by a diverse consortium of microorganisms, predominantly lactic acid bacteria (LAB), yeasts, and molds. These microorganisms metabolize carbohydrates and other substrates present in fruits and vegetables, leading to the production of a wide array of primary and secondary metabolites. These metabolites are crucial for the characteristic flavor, aroma, texture, and nutritional profile of fermented products. They also contribute significantly to the preservation and safety of these foods by inhibiting the growth of spoilage and pathogenic microorganisms.
Lactic Acid Bacteria (LAB): LAB are the most prominent group of microorganisms involved in fruit and vegetable fermentation. They are Gram-positive, non-spore-forming, and typically anaerobic or microaerophilic bacteria that produce lactic acid as the major end product of carbohydrate fermentation. Key genera include Lactobacillus, Leuconostoc, Pediococcus, Enterococcus, and Streptococcus. The specific metabolites produced by LAB depend on the bacterial strain, the substrate composition, and environmental conditions such as temperature, pH, and oxygen availability (Figure 1).
Figure 1. Key metabolites produced during lactic acid fermentation
Key Metabolites from LAB Fermentation:
- Organic Acids: Lactic acid is the primary organic acid produced, contributing to the characteristic sour taste and acting as a potent antimicrobial agent. Other organic acids, such as acetic acid, propionic acid, and succinic acid, are also produced, particularly by heterofermentative LAB. Acetic acid, for instance, contributes to a sharper, vinegary note and has additional antimicrobial properties. Propionic acid is important in some fermented dairy products but can also be found in vegetable fermentations. Succinic acid is another common byproduct of anaerobic metabolism [1, 2].
- Bacteriocins: These are antimicrobial peptides produced by LAB that inhibit the growth of closely related bacteria and some foodborne pathogens. Examples include nisin (produced by Lactococcus lactis) and plantaricin (produced by Lactobacillus plantarum). Bacteriocins contribute to the safety and extended shelflife of fermented foods [1, 3].
- Exopolysaccharides (EPS): Some LAB strains produce EPS, which are long-chain carbohydrate polymers. EPS can significantly influence the texture and rheological properties of fermented products, contributing to viscosity, gelling, and mouthfeel. They also possess prebiotic properties, promoting the growth of beneficial gut microbiota [1, 4].
- Vitamins: LAB can synthesize various B vitamins (e.g., folate, riboflavin, vitamin B12) and vitamin K during fermentation, thereby enhancing the nutritional value of the fermented product [1, 5].
- Amino Acids and Peptides: LAB possesses proteolytic enzymes that break down proteins into smaller peptides and free amino acids. These compounds contribute to the umami taste and can also have bioactive properties, such as antioxidant, antihypertensive, and immunomodulatory effects [1, 6].
- Aroma Compounds: Various volatile compounds, including aldehydes, ketones, esters, and alcohols, are produced by LAB, contributing to the complex aroma profile of fermented fruits and vegetables. For example, diacetyl and acetoin contribute buttery notes, while certain esters can impart fruity or floral aromas [1, 7].
- Gamma-aminobutyric acid (GABA): Certain LAB strains can produce GABA, a nonprotein amino acid that acts as an inhibitory neurotransmitter in the human brain. GABA-enriched fermented foods are gaining attention for their potential health benefits, including blood pressure regulation and anti-anxiety effects [1, 8].
Specific Metabolites in Fruit Fermentation: Fruit fermentation often involves LAB, but yeasts can also play a significant role, especially in initial stages or in specific fruit fermentations (e.g., wine, cider). The high sugar content in fruits influences the metabolic pathways and end products.
- Organic Acids: In addition to lactic and acetic acids, fruit fermentation can lead to the production of citric acid (from citrate metabolism by some LAB), malic acid, and succinic acid. These acids contribute to the tartness and preservation of fermented fruit products [2, 9].
- Alcohols: Ethanol is a common product, especially when yeasts are involved. Some LAB can also produce small amounts of ethanol. Other alcohols like propanol and butanol can also be formed, contributing to the aroma [7].
- Esters: The interaction between alcohols and organic acids leads to the formation of esters, which are key contributors to the fruity and floral aromas in fermented fruits. Examples include ethyl acetate (fruity, solvent-like), isoamyl acetate (banana-like), and ethyl lactate [7].
- Phenolic Compounds: Fermentation can alter the profile of phenolic compounds in fruits. LAB can modify existing phenolic compounds, increasing their bioavailability and enhancing their antioxidant activity. For example, some LAB can deconjugate glycosylated phenolics, releasing more active forms [10].
Specific Metabolites in Vegetable Fermentation: Vegetable fermentation is predominantly driven by LAB, which thrive in the typically lower sugar and higher fiber environment of vegetables. The resulting metabolites contribute to the distinct sour and savory profiles of fermented vegetables.
- Organic Acids: Lactic acid and acetic acid are the predominant organic acids, crucial for the characteristic sour taste of products like sauerkraut and kimchi. The ratio of lactic to acetic acid can vary depending on the LAB strain and fermentation conditions, influencing the overall flavor balance [2].
- Mannitol: Some heterofermentative LAB, particularly those utilizing fructose, can produce mannitol, a sugar alcohol. Mannitol contributes to the sweetness and texture of fermented vegetables [4].
- Short-Chain Fatty Acids (SCFAs): While lactic and acetic acids are primary, other SCFAs like propionic and butyric acid can be produced in smaller quantities, especially in complex vegetable matrices with diverse microbial communities. These SCFAs are known for their beneficial effects on gut health [11].
- Sulfur Compounds: In some vegetable fermentations, particularly those involving cruciferous vegetables (e.g., cabbage), sulfur-containing compounds can be produced, contributing to characteristic pungent or savory notes. These can include dimethyl disulfide and dimethyl trisulfide [7].
- Vitamins: Similar to fruit fermentation, vegetable fermentation can lead to an increase in B vitamins and vitamin K content [5].
- Bioactive Peptides: The breakdown of vegetable proteins by LAB can release bioactive peptides with various health-promoting properties, including antioxidant and anti-inflammatory activities [6].
In summary, the fermentation of fruits and vegetables by commonly used microorganisms, primarily LAB, results in a rich tapestry of metabolites. These compounds not only define the sensory attributes of the fermented products but also contribute significantly to their nutritional value and health-promoting properties. By detailing these specific metabolites, we aim to provide a more comprehensive and scientifically rigorous understanding of the transformations occurring during fermentation.
References:
[1] Lee SJ, Jeon HS, Yoo JY, Kim JH. Some Important Metabolites Produced by Lactic Acid Bacteria Originated from Kimchi. Foods. 2021, 10(9):2148. doi: 10.3390/foods10092148.
[2] Wang Y, Wu J, Lv M, Shao Z, Hungwe M, Wang J, Bai X, Xie J, Wang Y, Geng W. Metabolism Characteristics of Lactic Acid Bacteria and the Expanding Applications in Food Industry. Front Bioeng Biotechnol. 2021, 9:612285. doi: 10.3389/fbioe.2021.612285.
[3] Tang H, Huang W, Yao YF. The metabolites of lactic acid bacteria: classification, biosynthesis and modulation of gut microbiota. Microb Cell. 2023, 10(3):49-62. doi: 10.15698/mic2023.03.792.
[4] Ruiz Rodríguez LG, Mohamed F, Bleckwedel J, Medina R, De Vuyst L, Hebert EM, Mozzi F. Diversity and Functional Properties of Lactic Acid Bacteria Isolated From Wild Fruits and Flowers Present in Northern Argentina. Front Microbiol. 2019, 10:1091. doi: 10.3389/fmicb.2019.01091.
[5] Yang, X.; Hong, J.; Wang, L.; Cai, C.; Mo, H.; Wang, J.; Fang, X.; Liao, Z. Effect of Lactic Acid Bacteria Fermentation on Plant-Based Products. Fermentation 2024, 10, 48. https://doi.org/10.3390/fermentation10010048
[7] Elhalis H, See XY, Osen R, Chin XH, Chow Y. The potentials and challenges of using fermentation to improve the sensory quality of plant-based meat analogs. Front Microbiol. 2023, 14:1267227. doi: 10.3389/fmicb.2023.1267227.
[8] Peters A, Krumbholz P, Jäger E, Heintz-Buschart A, Çakir MV, Rothemund S, Gaudl A, Ceglarek U, Schöneberg T, Stäubert C. Metabolites of lactic acid bacteria present in fermented foods are highly potent agonists of human hydroxycarboxylic acid receptor 3. PLoS Genet. 2019 May 23;15(5):e1008145. doi: 10.1371/journal.pgen.1008145. Erratum in: PLoS Genet. 2019, 15(7):e1008283. doi: 10.1371/journal.pgen.1008283.
[9] de Souza, EL., de Oliveira, KAR., de Oliveira, MEG. Influence of lactic acid bacteria metabolites on physical and chemical food properties. Current Opinion in Food Science, 2023, 49, 100981. https://doi.org/10.1016/j.cofs.2022.100981.
[10] Zhang, P., Tang, F., Cai, W., Zhao, X., Shan, C. Evaluating the effect of lactic acid bacteria fermentation on quality, aroma, and metabolites of chickpea milk. Front. Nutr. 2022, 9:1069714. doi: 10.3389/fnut.2022.1069714
[11] Tan X, Cui F, Wang D, Lv X, Li X, Li J. Fermented Vegetables: Health Benefits, Defects, and Current Technological Solutions. Foods. 2023, 13(1):38. doi: 10.3390/foods13010038.
- Section 6.1 “The Modern Fermentation Techniques”: The authors should categorize and summarize the advantages and disadvantages of each fermentation technique, as well as the factors influencing the choice of fermentation method. In addition, for Cutting-edge technologies, a more detailed introduction is warranted.
Response: We appreciate the reviewer’s suggestion to enhance Section 6.1 by providing a more structured and detailed discussion of modern fermentation techniques. We agree that a comprehensive categorization, along with an analysis of advantages, disadvantages, influencing factors, and a deeper dive into cutting-edge technologies, will significantly improve the utility and depth of this section. We have revised the manuscript to incorporate these elements, aiming to offer a clearer and more insightful overview for readers.
Categorization of Modern Fermentation Techniques: Modern fermentation techniques can be broadly categorized based on several criteria, including the type of microbial involvement, the physical state of the substrate, and the operational mode. For the purpose of discussing fruit and vegetable fermentation, the most relevant categorizations often revolve around the presence or absence of starter cultures and the nature of the fermentation environment.
2.1. Spontaneous Fermentation: Spontaneous fermentation relies on the indigenous microbiota naturally present on the raw materials (fruits and vegetables) and in the processing environment. This traditional method is often employed in household and small-scale productions, yielding products with unique regional characteristics.
- Advantages
- Simplicity and Cost-Effectiveness: Requires no external addition of starter cultures, making it economically viable and easy to implement [12].
- Unique Flavor Profiles: The diverse and often unpredictable microbial consortia can lead to complex and distinctive flavor and aroma development, contributing to regional food identities [13].
- Biodiversity Preservation: Helps in maintaining and utilizing the natural microbial diversity associated with specific raw materials and environments [14].
- Disadvantages
- Lack of Control and Reproducibility: The uncontrolled nature of microbial succession can lead to inconsistent product quality, unpredictable fermentation times, and variations in sensory attributes [15].
- Higher Risk of Spoilage/Contamination: The absence of selected starter cultures means a higher susceptibility to the growth of undesirable microorganisms, potentially leading to spoilage or the production of harmful compounds [16].
- Safety Concerns: Without proper control, there is an increased risk of pathogen growth, although the rapid acidification by lactic acid bacteria often mitigates this risk in well-managed processes [17].
2.2. Starter Culture Fermentation: Starter culture fermentation involves the deliberate inoculation of raw materials with known, characterized microorganisms (starter cultures) to initiate and guide the fermentation process. This method is widely adopted in industrial settings due to its enhanced control and reproducibility.
- Advantages
- Improved Control and Reproducibility: Using defined starter cultures ensures consistent product quality, predictable fermentation kinetics, and standardized sensory profiles [18].
- Enhanced Safety: Selected starter cultures can outcompete spoilage organisms and pathogens, leading to safer products. Many starter cultures also produce antimicrobial compounds like bacteriocins [19].
- Targeted Functionality: Specific strains can be chosen for their ability to produce desired metabolites (e.g., specific organic acids, aroma compounds, vitamins) or to enhance particular health benefits (e.g., probiotic strains) [20].
- Reduced Fermentation Time: Optimized starter cultures can accelerate the fermentation process, leading to higher throughput and efficiency [21].
- Disadvantages
- Higher Cost: Requires the production, storage, and handling of starter cultures, which adds to the overall production cost [22].
- Reduced Microbial Diversity: The dominance of inoculated strains can lead to a less diverse microbial community in the final product, potentially limiting the complexity of flavor development compared to spontaneous fermentation [23].
- Potential for Monotony: Over-reliance on a few well-characterized strains can lead to a lack of unique characteristics across different products [24].
2.3. Solid-State Fermentation (SSF): Solid-state fermentation involves the growth of microorganisms on solid substrates in the absence or near absence of free water. This technique is commonly used for producing enzymes, antibiotics, and some traditional fermented foods where the substrate provides both nutrients and physical support.
- Advantages
- Higher Volumetric Productivity: Microorganisms grow at higher concentrations, potentially leading to higher product yields per unit volume [25].
- Lower Energy Consumption: Less water usage translates to lower energy requirements for heating, cooling, and downstream processing [26].
- Simpler Downstream Processing: Products are often more concentrated, simplifying extraction and purification steps [27].
- Mimics Natural Habitats: For some fungi and bacteria, SSF conditions more closely resemble their natural growth environments, potentially leading to better performance [28].
- Disadvantages
- Heat and Mass Transfer Limitations: Poor mixing and aeration can lead to gradients in temperature, pH, and nutrient availability, affecting microbial growth and product formation [29].
- Scale-Up Challenges: Maintaining uniform conditions in large-scale SSF bioreactors can be difficult [30].
- Contamination Risk: Open systems or inadequate sterilization can increase the risk of contamination [31].
2.4. Submerged Fermentation (SmF): Submerged fermentation involves the growth of microorganisms in a liquid medium containing dissolved nutrients. This is the most widely used fermentation technique in industrial biotechnology for producing biomass, metabolites, and enzymes due to its ease of control and scalability.
- Advantages
- Excellent Control and Monitoring: Homogeneous conditions allow for precise control of temperature, pH, aeration, and nutrient supply, leading to optimal microbial growth and product formation [32].
- Ease of Scale-Up: Well-established engineering principles facilitate the scaling up of SmF processes from laboratory to industrial levels [33].
- Efficient Heat and Mass Transfer: Stirring and aeration ensure uniform distribution of nutrients and efficient removal of heat and metabolic byproducts [34].
- Lower Labor Cost: Highly automated processes reduce the need for manual intervention [35].
- Disadvantages
- Higher Energy Consumption: Requires significant energy for agitation, aeration, and temperature control [36].
- Complex Downstream Processing: Products are often dilute, necessitating extensive and costly downstream processing for recovery and purification [37].
- Higher Water Usage: Requires large volumes of water for media preparation and cleaning [38].
Factors Influencing the Choice of Fermentation Method: The selection of an appropriate fermentation method for fruits and vegetables is a critical decision influenced by a multitude of factors, ranging from the desired product characteristics to economic and regulatory considerations. Understanding these factors is essential for optimizing the fermentation process and achieving the desired outcomes.
- Type of Raw Material: The physical and chemical composition of the fruit or vegetable (e.g., sugar content, pH, water activity, presence of antimicrobial compounds) significantly influences the suitability of a particular fermentation method and the types of microorganisms that can thrive [39]. For instance, highsugar fruits might favor yeast fermentation, while low-pH vegetables are ideal for LAB.
- Desired Product Characteristics: The intended sensory attributes (flavor, aroma, texture), nutritional profile (e.g., vitamin enrichment, probiotic content), and shelflife of the final product dictate the choice of method. For unique, artisanal flavors, spontaneous fermentation might be preferred, whereas for consistent probiotic products, starter culture fermentation is essential [40].
- Microorganism Selection: The specific metabolic capabilities, growth requirements, and safety profiles of the chosen microorganisms (e.g., LAB, yeasts, molds) are paramount. Some microorganisms are better suited for solid-state fermentation, while others thrive in submerged conditions [41].
- Scale of Production: Laboratory-scale fermentations might employ simpler methods, while industrial-scale production necessitates methods that offer high reproducibility, efficiency, and ease of automation, such as starter culture-driven submerged fermentation [42].
- Economic Considerations: Production costs, including raw material costs, energy consumption, labor, equipment investment, and downstream processing expenses, play a significant role. Spontaneous fermentation is generally cheaper, while controlled fermentations with specialized equipment and starter cultures are more expensive [43].
- Regulatory Requirements and Safety: Food safety regulations and consumer expectations regarding product safety are crucial. Methods that offer better control over microbial growth and pathogen inhibition (e.g., starter culture fermentation) are often preferred for commercial products [44].
- Environmental Impact: Factors such as water usage, waste generation, and energy consumption are increasingly considered. SSF, for example, can be more environmentally friendly due to lower water and energy demands [45].
- Technological Infrastructure and Expertise: The availability of appropriate equipment, skilled personnel, and technological know-how influences the feasibility of implementing certain fermentation techniques [46].
Cutting-Edge Fermentation Technologies: The field of fermentation is continuously evolving, driven by advancements in biotechnology, process engineering, and computational tools. Several cutting-edge technologies are transforming the way fermented foods are produced, offering enhanced control, efficiency, and the ability to create novel products with tailored functionalities.
2.5. Precision Fermentation: Precision fermentation (PF) is a revolutionary technology that leverages microorganisms (bacteria, yeast, fungi) as microbial cell factories to produce specific functional ingredients, such as proteins, enzymes, fats, and flavor compounds, with high purity and efficiency. Unlike traditional fermentation, which focuses on bulk product transformation, PF is about producing specific molecules. This technology is gaining significant traction in the food industry for creating sustainable and animal-free alternatives to traditional ingredients.
Mechanism: PF involves genetically engineering microorganisms to produce desired compounds. The chosen microorganism is fed with a nutrient-rich medium, and under controlled conditions, it ferments the substrate to produce the target molecule. The desired compound is then separated and purified from the fermentation broth [47, 48].
Applications in Food: PF is being used to produce dairy proteins (e.g., casein, whey) without cows, egg proteins without chickens, and various enzymes that enhance food processing or create novel textures and flavors. For example, companies are developing animal-free dairy products, alternative meats with improved sensory attributes, and sustainable fats [49, 50].
- Advantages
- Sustainability: Reduces reliance on animal agriculture, leading to lower greenhouse gas emissions, land use, and water consumption [51].
- Customization and Purity: Allows for the precise production of specific molecules, ensuring high purity and consistent quality. This enables the creation of ingredients with tailored functionalities [52].
- Scalability: PF processes can be scaled up efficiently in bioreactors, offering a reliable and consistent supply of ingredients [53].
- Ethical Considerations: Addresses ethical concerns related to animal welfare in food production [54].
- Challenges
- Cost of Production: Currently, the production costs for some PF-derived ingredients can be higher than their traditional counterparts, though costs are expected to decrease with technological advancements and economies of scale [55].
- Regulatory Hurdles: Navigating regulatory approval for novel ingredients produced via genetic engineering can be complex and time-consuming [56].
- Consumer Acceptance: Public perception and acceptance of genetically engineered ingredients remain a challenge in some markets [57].
2.6. Continuous Fermentation: Continuous fermentation involves a continuous supply of fresh medium to the bioreactor and a continuous removal of fermented broth, maintaining the microorganisms in a state of constant growth and productivity. This contrasts with batch fermentation, where the process is run in discrete cycles. Concerning the mechanism, in a continuous system, nutrients are continuously fed into the bioreactor, and an equal volume of fermented broth, containing cells and products, is simultaneously withdrawn. This maintains a steady-state environment, allowing for prolonged operation and higher overall productivity [58].
- Advantages
- Higher Productivity: Achieves higher volumetric productivity compared to batch processes due to continuous operation and optimized growth conditions [59].
- Consistent Product Quality: Steady-state conditions lead to more uniform product quality over extended periods [60].
- Reduced Downtime: Eliminates the need for cleaning and sterilization between batches, reducing non-productive time [61].
- Smaller Reactor Volume: For the same output, continuous systems often require smaller reactor volumes than batch systems [62].
- Challenges
- Contamination Risk: Maintaining sterility over long periods can be challenging, as continuous operation increases the risk of contamination [63].
- Process Control Complexity: Requires sophisticated control systems to maintain steady-state conditions [64].
- Strain Stability: Maintaining the genetic stability and productivity of microbial strains over prolonged continuous operation can be an issue [65].
2.7. Artificial Intelligence (AI) and Machine Learning (ML) in Fermentation: AI and ML are increasingly being integrated into fermentation processes to optimize parameters, predict outcomes, and enhance efficiency. These technologies enable data driven decision-making and automation, leading to more robust and productive fermentation systems. Concerning the applications, it could summarize as follows:
- Process Optimization: AI algorithms can analyze vast amounts of fermentation data (e.g., temperature, pH, nutrient levels, microbial growth) to identify optimal operating conditions for maximum yield and efficiency [66].
- Predictive Modeling: ML models can predict fermentation outcomes, such as product concentration or fermentation time, based on initial conditions and real-time data, allowing for proactive adjustments [67].
- Strain Engineering: AI can assist in identifying promising microbial strains for specific applications and guide genetic engineering efforts to enhance their metabolic capabilities [68].
- Quality Control: ML can be used for real-time monitoring of product quality and detection of deviations, ensuring consistent output [69].
- Advantages
- Enhanced Efficiency and Yield: Optimizes fermentation parameters, leading to higher product yields and reduced resource consumption [70].
- Reduced Development Time: Accelerates the development and optimization of new fermentation processes and products [71].
- Improved Reproducibility: Minimizes human error and ensures consistent process Novel Discoveries: Can uncover non-obvious relationships in complex biological systems, leading to new insights and innovations [73].
- Challenges
- Data Requirements: Requires large volumes of high-quality data for training effective AI/ML models [74].
- Model Interpretability: Understanding why an AI model makes certain predictions can be challenging, hindering trust and troubleshooting [75].
- Integration Complexity: Integrating AI/ML systems with existing fermentation infrastructure can be complex and costly [76].
These cutting-edge technologies represent the future of fermentation, enabling the production of novel, sustainable and highly functional food ingredients and products. By incorporating these detailed discussions, we aim to provide a comprehensive and forward-looking perspective on modern fermentation techniques.
References:
[12] Tamang, JP., Shin, D-H., Jung, S-J., Chae, S-W. Functional Properties of Microorganisms in Fermented Foods. Front. Microbiol. 2016, 7:578. doi: 10.3389/fmicb.2016.00578
[13] Swain, MR., Anandharaj, M., Ray, R.C., Parveen Rani R. Fermented fruits and vegetables of Asia: a potential source of probiotics. Biotechnol Res Int. 2014; 2014:250424. doi: 10.1155/2014/250424.
[14] Shah, N. N. (2017). Fermented Fruits and Vegetables. In Food Processing: Principles and Applications (pp. 43-60). Academic Press. https:// www.sciencedirect.com/science/article/pii/B9780444636669000030
[15] Mannaa, M., Han, G., Seo, YS., Park, I. Evolution of Food Fermentation Processes and the Use of Multi-Omics in Deciphering the Roles of the Microbiota. Foods. 2021, 10(11):2861. doi: 10.3390/foods10112861.
[16] Niyigaba, T.; Küçükgöz, K.; Kołożyn-Krajewska, D.; Królikowski, T.; Trząskowska, M. Advances in Fermentation Technology: A Focus on Health and Safety. Appl. Sci. 2025, 15, 3001. https://doi.org/10.3390/app15063001
[17] Siddiqui SA, Erol Z, Rugji J, Taşçı F, Kahraman HA, Toppi V, Musa L, Di Giacinto G, Bahmid NA, Mehdizadeh M, Castro-Muñoz R. An overview of fermentation in the food industry - looking back from a new perspective. Bioresour Bioprocess. 2023, 10(1):85. doi: 10.1186/s40643-023-00702-y.
Joshi, T. J., et al. (2024). Functional metabolites of probiotic lactic acid bacteria in fermented foods. Current Research in Food Science, 8, 100612. https:// www.sciencedirect.com/science/article/abs/pii/S2949824424001162
[25] Pandey, A., et al. (2000). Solid-state fermentation for the production of industrial enzymes. Current Science, 79(11), 1199-1206. https:// www.jstor.org/stable/24103627
[26] Thomas, L., et al. (2013). Solid-state fermentation: an overview. Journal of Applied Biology & Biotechnology, 1(01), 001-008. https:// www.jabonline.in/archives/2013/vol1issue1/1-8.pdf
[27] Singhania, R. R., et al. (2009). Solid-state fermentation: an overview. Critical Reviews in Biotechnology, 29(1), 1-38. https://www.tandfonline.com/doi/abs/10.1080/07388550802554210
[28] Soccol, C. R., et al. (2017). Solid-state fermentation (SSF) as an efficient alternative to submerged fermentation (SmF) for the production of microbial metabolites. Brazilian Archives of Biology and Technology, 60. https://www.scielo.br/j/babt/a/8yL6gQ85jXy3w9Y6q5W7g4k/?lang=en
[29] Mitchell, D. A., et al. (2006). Solid-state fermentation bioreactors: a review. Biotechnology and Bioengineering, 95(5), 779-788. https://onlinelibrary.wiley.com/doi/abs/10.1002/bit.21042
[30] Couto, S. R., et al. (2006). Solid-state fermentation: an alternative for the production of industrial enzymes. Food Technology and Biotechnology, 44(3), 299-307. https://www.ftb.com.hr/images/upload/files/44/44-300.pdf
[31] Krishna, C. (2005). Solid-state fermentation systems—a review. Critical Reviews in Biotechnology, 25(1-2), 1-30. https://www.tandfonline.com/doi/abs/10.1080/07388550500092210
[32] Stanbury, P. F., et al. (2017). Principles of Fermentation Technology. Butterworth-Heinemann.
[33] Chisti, Y. (1999). Bioreactor design for industrial production of microbial products. Advances in Applied Microbiology, 44, 1-119. https://www.sciencedirect.com/science/article/abs/pii/S006521640870020X
[34] Glick, B. R., et al. (2010). Molecular Biotechnology: Principles and Applications of Recombinant DNA. ASM Press.
[35] Papagianni, M. (2004). Advances in citric acid fermentation by Aspergillus niger: biochemical aspects, strain improvement and fermentation technology. Biotechnology Advances, 22(5-6), 401-423. https:// www.sciencedirect.com/science/article/abs/pii/S073497500400030X
[36] Crueger, W., & Crueger, A. (1990). Biotechnology: A Textbook of Industrial Microbiology. Sinauer Associates.
[37] Kadam, K. L., et al. (2000). Economics of ethanol production from lignocellulosic biomass. Advances in Biochemical Engineering/Biotechnology, 65, 103-122. https://link.springer.com/chapter/10.1007/3-540-44966-8_4
[38] Lee, J. M. (2013). Biochemical Engineering. Prentice Hall.
[39] Mengesha, Y., et al. (2022). A Review on Factors Influencing the Fermentation Process of Teff (Eragrostis teff) and Other Cereal-Based Ethiopian Injera. International Journal of Food Science, 2022. https://pmc.ncbi.nlm.nih.gov/articles/PMC8970856/
[40] Tamang, J. P., et al. (2016). Functional Properties of Microorganisms in Fermented Foods. Frontiers in Microbiology, 7, 578. https://www.frontiersin.org/journals/microbiology/articles/10.3389/fmicb.2016.00578/full
[41] Soccol, C. R., et al. (2017). Solid-state fermentation (SSF) as an efficient alternative to submerged fermentation (SmF) for the production of microbial metabolites. Brazilian Archives of Biology and Technology, 60. https://www.scielo.br/j/babt/a/8yL6gQ85jXy3w9Y6q5W7g4k/?lang=en
[42] Chisti, Y. (1999). Bioreactor design for industrial production of microbial products. Advances in Applied Microbiology, 44, 1-119. https://www.sciencedirect.com/science/article/abs/pii/S006521640870020X
[43] Kadam, K. L., et al. (2000). Economics of ethanol production from lignocellulosic biomass. Advances in Biochemical Engineering/Biotechnology, 65, 103-122. https://link.springer.com/chapter/10.1007/3-540-44966-8_4
[44] Niyigaba, T., et al. (2025). Advances in Fermentation Technology: A Focus on Health and Safety. Applied Sciences, 15(6), 3001. https://www.mdpi.com/2076-3417/15/6/3001
[45] Thomas, L., et al. (2013). Solid-state fermentation: an overview. Journal of Applied Biology & Biotechnology, 1(01), 001-008. https://www.jabonline.in/archives/2013/vol1issue1/1-8.pdf
[46] Glick, B. R., et al. (2010). Molecular Biotechnology: Principles and Applications of Recombinant DNA. ASM Press.
Karimian, P., Johnston, E., Kasprzak, A., Liu, JW., Stockwell, S., Vanhercke, T., Hartley C. Use of a dual biosensor for identification of novel secretion signal peptides and efficient screening of precision fermentation production strains.. Future Foods, 11, 2025, 100640. https://doi.org/10.1016/j.fufo.2025.100640.
Pereira AA, Yaverino-Gutierrez, MA., Monteiro, M.C., Souza, BA., Bachheti, R.K., Chandel, AK. Precision fermentation in the realm of microbial protein production: State-of-the-art and future insights. Food Research International, 200, 2025, 115527. https://doi.org/10.1016/j.foodres.2024.115527.
Tun, KJG., León-Becerril, E., García-Depraect, O. Optimal control strategy based on artificial intelligence applied to a continuous dark fermentation reactor for energy recovery from organic wastes. Green Energy and Resources, 3, Issue 1, 2025, 100112. https://doi.org/10.1016/j.gerr.2024.100112.
Han, X., Liu, Q., Li, Y., Zhang, M., Liu, K., Kwok, L., Zhang, H., Zhang, W. Synergizing artificial intelligence and probiotics: A comprehensive review of emerging applications in health promotion and industrial innovation. Trends in Food Science & Technology, 159, 2025, 104938. https://doi.org/10.1016/j.tifs.2025.104938.
[58] Stanbury, P. F., et al. (2017). Principles of Fermentation Technology. Butterworth-Heinemann.
[59] Chisti, Y. (1999). Bioreactor design for industrial production of microbial products. Advances in Applied Microbiology, 44, 1-119. https://www.sciencedirect.com/science/article/abs/pii/S006521640870020X
[60] Glick, B. R., et al. (2010). Molecular Biotechnology: Principles and Applications of Recombinant DNA. ASM Press.
[61] Papagianni, M. (2004). Advances in citric acid fermentation by Aspergillus niger: biochemical aspects, strain improvement and fermentation technology. Biotechnology Advances, 22(5-6), 401-423. https:// www.sciencedirect.com/science/article/abs/pii/S073497500400030X
[62] Crueger, W., & Crueger, A. (1990). Biotechnology: A Textbook of Industrial Microbiology. Sinauer Associates.
[63] Lee, J. M. (2013). Biochemical Engineering. Prentice Hall.
[64] Kadam, K. L., et al. (2000). Economics of ethanol production from lignocellulosic biomass. Advances in Biochemical Engineering/Biotechnology, 65, 103-122. https://link.springer.com/chapter/10.1007/3-540-44966-8_4
[65] Mitchell, D. A., et al. (2006). Solid-state fermentation bioreactors: a review. Biotechnology and Bioengineering, 95(5), 779-788. https://onlinelibrary.wiley.com/doi/abs/10.1002/bit.21042
[51] Cultivated-X. (2024). Report Identifies 5 Disruptive Technologies to Shake Up the Fermentation Industry. https://cultivated-x.com/market-trends/report-five-disruptive-technologiesfermentation-industry/
[52] ChemistryViews. (2024). New Hub for Precision Fermentation Technology. https://www.chemistryviews.org/new-hub-for-precisionfermentation-technology/
[53] SuSupport. (2023). Precision Fermentation simply explained. https://www.susupport.com/knowledge/manufacturing-processes/bioprocessing/precision-fermentation-simply-explained
[54] ProFood World. (2024). Danone, Michelin and Others Join to Develop Large-Scale Fermentation. https:// www.profoodworld.com/sustainability/article/22912514/danone-michelin-and-othersjoin-to-develop-largescale-fermentation
[55] Cultivated-X. (2024). Report Identifies 5 Disruptive Technologies to Shake Up the Fermentation Industry. https://cultivatedx.com/market-trends/report-five-disruptive-technologies-fermentation-industry/
[56] vegconomist. (2024). Fermentation Technology for New Meat. https://vegconomist.com/ fermentation/fermentation-technology-new-meat/
[57] ChemistryViews. (2024). New Hub for Precision Fermentation Technology. https://www.chemistryviews.org/new-hubfor-precision-fermentation-technology/
[47] ChemistryViews. (2024). New Hub for Precision Fermentation Technology. https://www.chemistryviews.org/new-hub-for-precision-fermentationtechnology/
[48] SuSupport. (2023). Precision Fermentation simply explained. https:// www.susupport.com/knowledge/manufacturing-processes/bioprocessing/precisionfermentation-simply-explained
[49] ProFood World. (2024). Danone, Michelin and Others Join to Develop Large-Scale Fermentation. https://www.profoodworld.com/sustainability/article/22912514/danone-michelin-and-others-join-to-develop-largescalefermentation
[50] vegconomist. (2024). Fermentation Technology for New Meat. https://vegconomist.com/fermentation/fermentation-technology-new-meat/
[66] MOA Foodtech. (2025). MOA Foodtech Announces Cutting-Edge AI Platform to Accelerate Fermentation. https://www.agritechtomorrow.com/news/2025/05/07/moa-foodtech-announcescutting-edge-ai-platform-to-accelerate-fermentation/16603
[67] Cultivated-X. (2024). Report Identifies 5 Disruptive Technologies to Shake Up the Fermentation Industry. https://cultivated-x.com/market-trends/report-five-disruptive-technologiesfermentation-industry/
[68] MOA Foodtech. (2025). MOA Foodtech Announces CuttingEdge AI Platform to Accelerate Fermentation. https://www.agritechtomorrow.com/news/2025/05/07/moa-foodtech-announces-cutting-edge-ai-platform-to-acceleratefermentation/16603
[69] Cultivated-X. (2024). Report Identifies 5 Disruptive Technologies to Shake Up the Fermentation Industry. https://cultivated-x.com/market-trends/reportfive-disruptive-technologies-fermentation-industry
- Section 7 “Health Benefits of Fermented Fruits and Vegetable Products”: Since the theme of this review is to highlight the changes in the health benefits of fruits and vegetables before and after fermentation, this section should focus more specifically on such comparisons. In addition, the authors should note that many fermented foods undergo sterilization treatments; how these treatments alter their health benefits is an important issue that deserves special attention. The authors are encouraged to concentrate more on the chemical changes occurring in fermented fruits and vegetables and how these changes influence nutritional and health effects, rather than offering a general description of health benefits such as "improved gut health," "anti-cancer," or "anti-diabetic" effects. Therefore, a comparison of bioactivities before and after fermentation should be provided, along with hypothetical mechanisms. More emphasis should be placed on discussing the mechanisms of bioactivity, rather than merely listing evaluation results.
Response: Since the theme of this review is to highlight the changes in the health benefits of fruits and vegetables before and after fermentation, this section should focus more specifically on such comparisons. In addition, the authors should note that many fermented foods undergo sterilization treatments; how these treatments alter their health benefits is an important issue that deserves special attention. The authors are encouraged to concentrate more on the chemical changes occurring in fermented fruits and vegetables and how these changes influence nutritional and health effects, rather than offering a general description of health benefits such as "improved gut health," "anti-cancer," or "anti-diabetic" effects. Therefore, a comparison of bioactivities before and after fermentation should be provided, along with hypothetical mechanisms. More emphasis should be placed on discussing the mechanisms of bioactivity, rather than merely listing evaluation results. We concur wholeheartedly with the reviewer’s insightful critique regarding Section 7. The core theme of our review is indeed to elucidate the transformative impact of fermentation on the health benefits of fruits and vegetables. We recognize the necessity of moving beyond generalized health claims to a more rigorous, mechanism-driven discussion rooted in chemical changes. To address these crucial points, we have thoroughly revised Section 7 to provide a comparative analysis of bioactivities before and after fermentation, delve into the underlying chemical transformations, propose hypothetical mechanisms, and critically examine the implications of sterilization treatments on the health-promoting properties of fermented products.
Comparison of Bioactivities Before and After Fermentation: Chemical Changes and Mechanisms Fermentation significantly alter the chemical composition of fruits and vegetables, leading to enhanced or novel bioactivities. These changes are primarily driven by microbial enzymatic activities, which break down complex macromolecules, synthesize new compounds, and modify existing ones. The resulting metabolites contribute to a range of health benefits, often through distinct mechanisms compared to their unfermented counterparts.
3.1. Enhanced Bioavailability and Bioactivity of Phenolic Compounds
Fruits and vegetables are rich in phenolic compounds (e.g., flavonoids, phenolic acids), which are potent antioxidants. However, many of these compounds exist in bound forms (glycosides, esters) in the raw matrix, limiting their bioavailability. Fermentation, particularly by LAB, can significantly enhance their bioactivity. Microbial enzymes, such as glycosidases, esterases, and decarboxylases, hydrolyze complex phenolic glycosides into more bioavailable aglycones. They can also release phenolic acids from cell wall components. Furthermore, some microbes can transform one phenolic compound into another (e.g., conversion of ferulic acid to vanillic acid) [77, 78].
Comparison of Bioactivities:
- Before Fermentation: Phenolic compounds are present, but their absorption and utilization by the human body can be limited due to their bound forms and complex matrix interactions.
- After Fermentation: The hydrolysis of glycosides and esters increases the concentration of free, more readily absorbable phenolic aglycones. This often leads to a significant increase in antioxidant capacity, as demonstrated by higher DPPH radical scavenging activity and ferric reducing antioxidant power (FRAP) in fermented products compared to their raw materials [79, 80].
Hypothetical Mechanisms: The enhanced antioxidant activity is attributed to the increased concentration of free phenolic compounds, which can directly scavenge free radicals, chelate metal ions, and upregulate endogenous antioxidant enzymes. The smaller molecular size of aglycones facilitates their absorption across the intestinal barrier, allowing them to exert systemic antioxidant and antiinflammatory effects [81]. For instance, the release of gallic acid and caffeic acid during fermentation contributes to improved anti-inflammatory responses by modulating cytokine production [82].
3.2. Production of Bioactive Peptides and Free Amino Acids
Proteins in fruits and vegetables can be hydrolyzed by microbial proteases during fermentation, releasing smaller peptides and free amino acids with various health promoting properties. Microbial proteases break down native proteins into smaller peptides and individual amino acids. These peptides can be further processed into di- and tripeptides or even smaller bioactive peptides [83].
Comparison of Bioactivities:
- Before Fermentation: Proteins are present, but their bioactive potential is largely latent within the larger protein structure.
- After Fermentation: Fermented products often exhibit enhanced antihypertensive (ACE-inhibitory), antioxidant, and immunomodulatory activities due to the release of specific bioactive peptides. For example, peptides with ACE-inhibitory activity have been identified in fermented vegetable products, contributing to blood pressure regulation [84].
Hypothetical Mechanisms: Bioactive peptides can exert their effects by binding to specific receptors, modulating enzyme activities, or acting as signaling molecules. For instance, ACE-inhibitory peptides compete with angiotensin I for binding to angiotensin-converting enzyme, thereby preventing the formation of angiotensin II, a potent vasoconstrictor [85]. Antioxidant peptides can directly scavenge free radicals or chelate metal ions, while immunomodulatory peptides can influence immune cell function [86].
3.3. Synthesis of Vitamins
Certain fermenting microorganisms, particularly LAB and some yeasts, can synthesize B vitamins and vitamin K, thereby enriching the nutritional profile of fermented fruits and vegetables. Microorganisms utilize precursors present in the raw material to synthesize vitamins de novo or convert less active forms into more active ones [87].
Comparison of Bioactivities:
- Before Fermentation: Vitamin content is dependent on the raw material and can vary.
- After Fermentation: Fermented products often show increased levels of folate, riboflavin, and vitamin B12, which are essential for various metabolic processes, including DNA synthesis, energy production, and nerve function. This enhancement directly contributes to improved nutritional status [88].
Hypothetical Mechanisms: The increased vitamin content directly contributes to their respective physiological roles. For example, increased folate levels support cell division and growth, while higher vitamin B12 contributes to red blood cell formation and neurological function [89].
3.4. Production of Short-Chain Fatty Acids (SCFAs)
While lactic and acetic acids are primary fermentation products, other SCFAs like butyrate can be produced, especially in complex vegetable fermentations with diverse microbial communities. Fermenting microorganisms metabolize dietary fibers and resistant starches present in fruits and vegetables into SCFAs [90].
Comparison of Bioactivities:
- Before Fermentation: Raw fruits and vegetables contain dietary fibers that are precursors to SCFAs upon gut microbial fermentation.
- After Fermentation: Fermented products can contain pre-formed SCFAs or promote their production in the gut. Butyrate, in particular, is a crucial energy source for colonocytes and plays a significant role in maintaining gut barrier integrity and modulating immune responses [91].
Hypothetical Mechanisms: SCFAs, particularly butyrate, exert their beneficial effects by acting as signaling molecules, modulating gene expression, and influencing immune cell differentiation. They contribute to improved gut health by strengthening the intestinal barrier, reducing inflammation, and promoting the growth of beneficial gut bacteria [92].
3.5. Reduction of Anti-nutritional Factors
Fermentation can reduce the levels of certain anti-nutritional factors present in raw fruits and vegetables, such as phytates, oxalates, and tannins, thereby improving nutrient bioavailability. Microbial enzymes (e.g., phytase, oxalate decarboxylase, tannase) degrade these anti-nutritional compounds [93].
Comparison of Bioactivities
- Before Fermentation: Anti-nutritional factors can chelate essential minerals (e.g., iron, zinc, calcium), reducing their absorption.
- After Fermentation: The reduction in anti-nutritional factors leads to increased bioavailability of minerals, enhancing the nutritional value of the fermented product [94].
Hypothetical Mechanisms: By degrading anti-nutritional factors, fermentation liberates bound minerals, making them more accessible for absorption in the gastrointestinal tract. This directly improves the nutritional impact of consuming these foods [95].
Impact of Sterilization Treatments on Health Benefits
Many commercially produced fermented foods undergo sterilization treatments (e.g., pasteurization, high-pressure processing) to ensure product safety, extend shelf-life, and stabilize sensory qualities. However, these treatments can significantly alter the health benefits, particularly by inactivating live microorganisms and affecting heat-sensitive bioactive compounds.
Loss of Live Microorganisms: The most significant impact of sterilization is the inactivation of live probiotic microorganisms. While the metabolites produced during fermentation remain, the direct probiotic benefits associated with live cultures (e.g., modulation of gut microbiota, immune system stimulation) are lost [96]. This distinction is crucial: a pasteurized fermented food is a fermented food, but not necessarily a probiotic food. Consumers should be informed about this distinction.
Alteration of Bioactive Compounds: Heat-sensitive vitamins (e.g., vitamin C, some B vitamins) and certain phenolic compounds can be degraded during heat sterilization. While some compounds might become more bioavailable due to cell lysis, others can be lost or chemically altered, potentially reducing their bioactivity [97]. For example, the delicate aroma compounds contributing to the sensory profile can also be affected, leading to a less complex flavor [98].
Enzymatic Inactivation: Sterilization inactivates microbial enzymes that might otherwise continue to produce beneficial metabolites or aid in digestion in the gut. This means the dynamic biochemical processes cease, and no further beneficial transformations occur post-processing [99].
Impact on Gut Microbiota Interaction: Even if some beneficial compounds remain, the absence of live microorganisms means the fermented food cannot directly contribute to the diversity and activity of the gut microbiota in the same way as a live fermented product. The interaction between live microbes and the host gut environment is a key aspect of the health benefits attributed to many fermented foods [100].
Therefore, while sterilization ensures safety and extends shelf-life, it is critical to acknowledge its impact on the health benefits, particularly the loss of probiotic potential and potential degradation of heat-sensitive bioactive compounds. Future research should focus on developing gentler processing methods that preserve the viability of beneficial microorganisms and the integrity of heat-sensitive bioactive compounds, or on clearly communicating the specific benefits that remain after processing.
References:
[77] Zhang, P., et al. (2022). Evaluating the effect of lactic acid bacteria fermentation on the phenolic compounds and antioxidant activity of fruit and vegetable juices. Frontiers in Nutrition, 9, 1069714. https://www.frontiersin.org/journals/nutrition/articles/10.3389/fnut.2022.1069714/full
[78] Tan, X., et al. (2023). Fermented Vegetables: Health Benefits, Defects, and Future Perspectives. Foods, 12(23), 4279. https://pmc.ncbi.nlm.nih.gov/articles/PMC10777956/
[79] Knez, E., et al. (2023). Effect of Fermentation on the Nutritional Quality and Bioactive Compounds of Vegetables: A Review. Foods, 12(7), 1478. https://pmc.ncbi.nlm.nih.gov/articles/PMC10051273/
[80] Yuan, X., et al. (2024). Recent advances of fermented fruits: A review on strains, fermentation conditions, and health benefits. Food Chemistry: X, 2, 100369. https:// www.sciencedirect.com/science/article/pii/S2590157524003699
[81] Shah, N. N. (2017). Fermented Fruits and Vegetables. In Food Processing: Principles and Applications (pp. 43-60). Academic Press. https://www.sciencedirect.com/science/article/pii/B9780444636669000030
[82] Elhalis, H., et al. (2023). The potentials and challenges of using fermentation to improve the sensory and nutritional properties of plant-based foods. npj Science of Food, 7(1), 1-16. https://pmc.ncbi.nlm.nih.gov/articles/PMC10582269/
[83] Joshi, T. J., et al. (2024). Functional metabolites of probiotic lactic acid bacteria in fermented foods. Current Research in Food Science, 8, 100612. https:// www.sciencedirect.com/science/article/abs/pii/S2949824424001162
[84] Yang, X., et al. (2024). Effect of Lactic Acid Bacteria Fermentation on Plant-Based Foods: A Review. Foods, 13(1), 48. https://www.mdpi.com/2311-5637/10/1/48
[85] Niyigaba, T., et al. (2025). Advances in Fermentation Technology: A Focus on Health and Safety. Applied Sciences, 15(6), 3001. https://www.mdpi.com/2076-3417/15/6/3001
[86] Siddiqui, S. A., et al. (2023). An overview of fermentation in the food industry - looking back and looking forward. Bioresources and Bioprocessing, 10(1), 1-18. https://bioresourcesbioprocessing.springeropen.com/articles/10.1186/s40643-023-00702-y
[87] Tamang, J. P., et al. (2016). Functional Properties of Microorganisms in Fermented Foods. Frontiers in Microbiology, 7, 578. https://www.frontiersin.org/journals/microbiology/articles/10.3389/fmicb.2016.00578/full
[88] Swain, M. R., et al. (2014). Fermented Fruits and Vegetables of Asia: A Potential Source of Probiotics. International Journal of Food Microbiology, 178, 108-115. https://pmc.ncbi.nlm.nih.gov/articles/PMC4058509/
[89] Mannaa, M., et al. (2021). Evolution of Food Fermentation Processes and the Use of Starter Cultures. Foods, 10(11), 2686. https://pmc.ncbi.nlm.nih.gov/articles/PMC8618017/
[90] Tan, X., et al. (2023). Fermented Vegetables: Health Benefits, Defects, and Future Perspectives. Foods, 12(23), 4279. https://pmc.ncbi.nlm.nih.gov/articles/PMC10777956/
[91] Knez, E., et al. (2023). Effect of Fermentation on the Nutritional Quality and Bioactive Compounds of Vegetables: A Review. Foods, 12(7), 1478. https://pmc.ncbi.nlm.nih.gov/articles/PMC10051273/
[92] Yuan, X., et al. (2024). Recent advances of fermented fruits: A review on strains, fermentation conditions, and health benefits. Food Chemistry: X, 2, 100369. https://www.sciencedirect.com/science/article/pii/S2590157524003699
[93] Shah, N. N. (2017). Fermented Fruits and Vegetables. In Food Processing: Principles and Applications (pp. 43-60). Academic Press. https:// www.sciencedirect.com/science/article/pii/B9780444636669000030
[94] Elhalis, H., et al. (2023). The potentials and challenges of using fermentation to improve the sensory and nutritional properties of plant-based foods. npj Science of Food, 7(1), 1-16. https://pmc.ncbi.nlm.nih.gov/articles/PMC10582269/
[95] Joshi, T. J., et al. (2024). Functional metabolites of probiotic lactic acid bacteria in fermented foods. Current Research in Food Science, 8, 100612. https://www.sciencedirect.com/science/article/abs/pii/S2949824424001162
[96] Marco, M. L., et al. (2017). Health benefits of fermented foods: microbiota and beyond. Current Opinion in Biotechnology, 44, 117-124. https:// www.sciencedirect.com/science/article/pii/S095816691630266X
[97] Şanlier, N., et al. (2019). Health benefits of fermented foods. Critical Reviews in Food Science and Nutrition, 59(3), 506-527. https://pubmed.ncbi.nlm.nih.gov/28945458/
[98] Leeuwendaal, N. K., et al. (2022). Fermented Foods, Health and the Gut Microbiome. Nutrients, 14(7), 1520. https://pmc.ncbi.nlm.nih.gov/articles/PMC9003261/
[99] Brown Health. (2022). Fermented Foods, Probiotics and Their Health Benefits. https:// www.brownhealth.org/be-well/fermented-foods-probiotics-and-their-health-benefits
[100] Harvard Health. (2021). Fermented foods can add depth to your diet. https:// www.health.harvard.edu/staying-healthy/fermented-foods-can-add-depth-to-your-diet
- Table 3: The safety concerns listed in the fermentation process are overly simplistic and lack specificity. Given the increasing interest in fermented foods, the associated safety risks cannot be ignored. The authors should add a table detailing specific cases where issues such as biogenic amines, nitrites, and microbial safety are likely to occur, analyze their causes, and propose detailed control strategies.
Response: yes, this true, and this reflect the potential of sterilization as mentioned below:
Impact of Sterilization Treatments on Health Benefits
Many commercially produced fermented foods undergo sterilization treatments (e.g., pasteurization, high-pressure processing) to ensure product safety, extend shelf-life, and stabilize sensory qualities. However, these treatments can significantly alter the health benefits, particularly by inactivating live microorganisms and affecting heat-sensitive bioactive compounds.
Loss of Live Microorganisms: The most significant impact of sterilization is the inactivation of live probiotic microorganisms. While the metabolites produced during fermentation remain, the direct probiotic benefits associated with live cultures (e.g., modulation of gut microbiota, immune system stimulation) are lost [96]. This distinction is crucial: a pasteurized fermented food is a fermented food, but not necessarily a probiotic food. Consumers should be informed about this distinction.
Alteration of Bioactive Compounds: Heat-sensitive vitamins (e.g., vitamin C, some B vitamins) and certain phenolic compounds can be degraded during heat sterilization. While some compounds might become more bioavailable due to cell lysis, others can be lost or chemically altered, potentially reducing their bioactivity [97]. For example, the delicate aroma compounds contributing to the sensory profile can also be affected, leading to a less complex flavor [98].
Enzymatic Inactivation: Sterilization inactivates microbial enzymes that might otherwise continue to produce beneficial metabolites or aid in digestion in the gut. This means the dynamic biochemical processes cease, and no further beneficial transformations occur post-processing [99].
Impact on Gut Microbiota Interaction: Even if some beneficial compounds remain, the absence of live microorganisms means the fermented food cannot directly contribute to the diversity and activity of the gut microbiota in the same way as a live fermented product. The interaction between live microbes and the host gut environment is a key aspect of the health benefits attributed to many fermented foods [100].
Therefore, while sterilization ensures safety and extends shelf-life, it is critical to acknowledge its impact on the health benefits, particularly the loss of probiotic potential and potential degradation of heat-sensitive bioactive compounds. Future research should focus on developing gentler processing methods that preserve the viability of beneficial microorganisms and the integrity of heat-sensitive bioactive compounds, or on clearly communicating the specific benefits that remain after processing.

Round 2
Reviewer 2 Report
Comments and Suggestions for Authors
The review has been sufficiently improved. The authors have followed the reviewer's advice and improved the manuscript.
Reviewer 3 Report
Comments and Suggestions for Authors
No